# The extracellular contractile injection system is enriched in environmental microbes and associates with numerous toxins

Alexander Martin Geller[1,3], Inbal Pollin[1,3], David Zlotkin[1], Aleks Danov[1], Nimrod Nachmias[1], William B. Andreopoulos[2], Keren Shemesh[1] & Asaf Levy [1✉]

The extracellular Contractile Injection System (eCIS) is a toxin-delivery particle that evolved from a bacteriophage tail. Four eCISs have previously been shown to mediate interactions between bacteria and their invertebrate hosts. Here, we identify eCIS loci in 1,249 bacterial and archaeal genomes and reveal an enrichment of these loci in environmental microbes and their apparent absence from mammalian pathogens. We show that 13 eCIS-associated toxin genes from diverse microbes can inhibit the growth of bacteria and/or yeast. We identify immunity genes that protect bacteria from self-intoxication, further supporting an anti-bacterial role for some eCISs. We also identify previously undescribed eCIS core genes, including a conserved eCIS transcriptional regulator. Finally, we present our data through an extensive eCIS repository, termed eCIStem. Our findings support eCIS as a toxin-delivery system that is widespread among environmental prokaryotes and likely mediates antagonistic interactions with eukaryotes and other prokaryotes.

[1] Department of Plant Pathology and Microbiology, the Robert H. Smith Faculty of Food and Environment, the Hebrew University of Jerusalem, Rehovot, Israel. [2] Department of Computer Science, San Jose State University, San Jose, CA, USA. [3] These authors contributed equally: Alexander Martin Geller, Inbal Pollin. ✉email: alevy@mail.huji.ac.il

The extracellular contractile injection system (eCIS; previously termed "PLTS", phage-like protein-translocation structures[1]) is a cell-free protein delivery system that is prevalent in bacteria and archaea, but its biological function is poorly understood. The eCIS particle resembles the contractile tail of a T4 bacteriophage and is mostly encoded by an operon of 15–28 genes[1]. This protein complex is 110–120 nm long and includes a baseplate, a sheathed hollow tube that has a needle-like tip (spike) on one side and a cap on the other side, and tail fibers that likely serve to adhere to target cells (Supplementary Fig. 1)[2,3]. eCIS contraction propels the tube out of the sheath, likely enabling the sharp tip to perforate the target cell membrane. Proteins that are injected by the particle eCIS into target cells upon contraction are called "effectors". These effectors are usually encoded at the 3′ end of the operon[4–6]. The effectors that have been studied were shown to perform enzymatic activities in the target eukaryotic cell, most of which lead to cell toxicity. eCIS shares structural similarity with other contractile nano weapons such as type VI secretion system (T6SS) and R-type pyocins but differs from these by being extracellular and by injecting effectors into the target cell instead of just perforating it, respectively. Despite the prevalence of eCISs across the microbial world, only four eCIS loci, Afp, AfpX, PVC, and MACs, have been experimentally studied.

The Antifeeding Prophage (Afp) is encoded by *Serratia entomophila* and it is sufficient to cause feeding cessation and death to the New Zealand grass grub pest[7–10]. A homologous Afp structure, termed AfpX, was described in *Serratia proteamaculans* and demonstrated insecticidal activity against the larvae of two insects[11]. Effectors of Afp and AfpX have not been experimentally confirmed. Photorhabdus Virulence Cassettes (PVC) was characterized in *Photorhabdus* spp. and has an injectable insecticidal activity against wax moth larvae[5]. Four effectors are associated with PVC and demonstrate cytotoxicity against eukaryotic cells: Plu1690[5], Pnf[4], RRSP$_{Pa}$[12], and SepC-like[5]. The high-resolution structures of Afp and PVC particles were recently solved using cryo-EM and provided excellent maps of the protein interactions within these particles[2,3]. Unlike the insecticidal activities of Afp, AfpX, and PVC, the metamorphosis-associated contractile structures (MACs) encoded by *Pseudoalteromonas luteoviolacea* plays a mutualistic role. MACs loci produce arrays of ~100 eCIS structures that trigger metamorphosis of the marine tubeworm *Hydroides elegans* hosting the bacteria[13]. MACs particles inject a nuclease effector that kills eukaryotic cell lines but this activity is not essential for metamorphosis[14]. Mif1 protein was shown as a cargo inside the MACs tube lumen but was not shown to act as a toxin[15]. To summarize, eCIS particles interact with different invertebrates and inject effectors that are toxic to eukaryotic cells. To the best of our knowledge, only seven eCIS effectors were experimentally validated as toxic to recipient cells or are known to be injected by eCIS. All known effectors are encoded in gammaproteobacteria.

Here we provide a systematic characterization of eCIS in the microbial world. We identified 1425 eCIS loci encoded within 1249 prokaryotic genomes, corresponding to 1.9% of the analyzed microbial genomes. Strikingly, eCIS loci are strongly enriched in environmental microbial taxa from different ecosystems and in microbiomes of specific hosts (plants, specific animals, protists), and are depleted from mammalian pathogens that have been extensively cultured and sequenced. We analyzed the proteins and protein domains of all eCIS loci and identified new core protein domains, including a putative metallopeptidase that associates with toxins and a putative master transcriptional regulator. We further bioinformatically analyzed the fiber proteins of eCIS that confer cell specificity and identified the first group of eCIS particles likely targeting bacteria. We identified a large set of putative toxins (which may play roles in nature as putative effectors) that are genetically associated with eCIS operons, the majority of which are specific to certain eCIS loci. Through heterologous expression experiments, we showed that 13 toxins were capable of killing *E. coli* and/or *S. cerevisiae* cells. In some cases, we were able to experimentally identify immunity genes that rescued bacteria from self-intoxication by their cognate toxins and detected a toxin that is more active in the periplasm, supporting the toxin intended activity against bacteria. Finally, we developed eCIStem; an online resource to present our findings that will serve as a valuable resource for the research community. We thus provide here an in-depth ecological and functional characterization of a poorly studied toxin-delivery system and its associated toxins.

## Results

### eCIS are encoded by 1.9% and 1.2% of sequenced bacteria and archaea, respectively, with a highly biased taxonomic distribution.

First, we were interested in identifying all eCIS loci in a large genomic dataset. We compiled a set of 64,756 microbial isolate genomes retrieved from Integrated Microbial Genomes (Supplementary Data 1)[16]. To identify core component homologs from known systems, we searched for genes with known eCIS-associated pfam annotations (Supplementary Table 1). To supplement this, we also annotated homologous genes ourselves by searching using the Hidden Markov Model (HMM) profiles from a recent publication[1,17]. We defined putative eCIS operons as gene cassettes that included these multiple eCIS core genes in close proximity and were not bacteriophage, T6SS, or R-type pyocins (Supplementary Table 1, Methods). Overall, we identified eCIS operons encoded in 1230 (1.9%) bacteria and 19 (1.2%) archaea from our genomic repository (Supplementary Data 2–3). We identified two core genes, *Afp8* and *Afp11*, that co-occur in eCIS operons across 98.7% of loci and used their protein sequences to construct an eCIS phylogenetic gene tree (Fig. 1a, Supplementary Figs. 2–3, Supplementary Data 4). Afp8 and Afp11 alone resulted in phylogenetically similar trees (Supplementary Fig. 4) and the trees agree with eCIS division into subtype I and II that were defined in a previous eCIS analysis[17] (Supplementary Fig. 5). eCIS is scattered across the prokaryotic diversity with presence in 14 bacterial phyla and one archaeal phylum. The incongruence between this tree and the genomic phylogeny suggests that eCIS undergo HGT frequently, as was proposed before[1,17]. The previously experimentally characterized eCISs are located within a narrow clade on the eCIS tree, pointing to the possibility that other eCIS particles may play more diverse ecological roles (Fig. 1a, Supplementary Fig. 2).

Next, we looked for genetic mechanisms that may mediate the eCIS HGT. Using Deeplasmid, a new plasmid prediction tool that we developed[18], we identified that 7.6% of eCIS are likely plasmid-borne (Fig. 1a and Supplementary Fig. 6, Supplementary Data 5, Methods). In other cases, we found a clear signature of eCIS operon integration into a specific bacterial chromosome (Supplementary Fig. 7). For example, we identified a likely homologous recombination event between identical tRNA genes, a classical integration site[19] (Supplementary Fig. 7b). These genomic integration events and the plasmid-borne eCIS operons shed light on the mechanisms through which eCIS loci have been horizontally propagated in microbial genomes.

### eCIS displays a highly biased taxonomic distribution.

Given the propensity of eCIS to transfer between microbes as phylogenetically distant as bacteria and archaea, we were surprised by its scarcity in microbial genomes. We tested if eCIS loci are homogeneously distributed across microbial taxa and found that eCIS

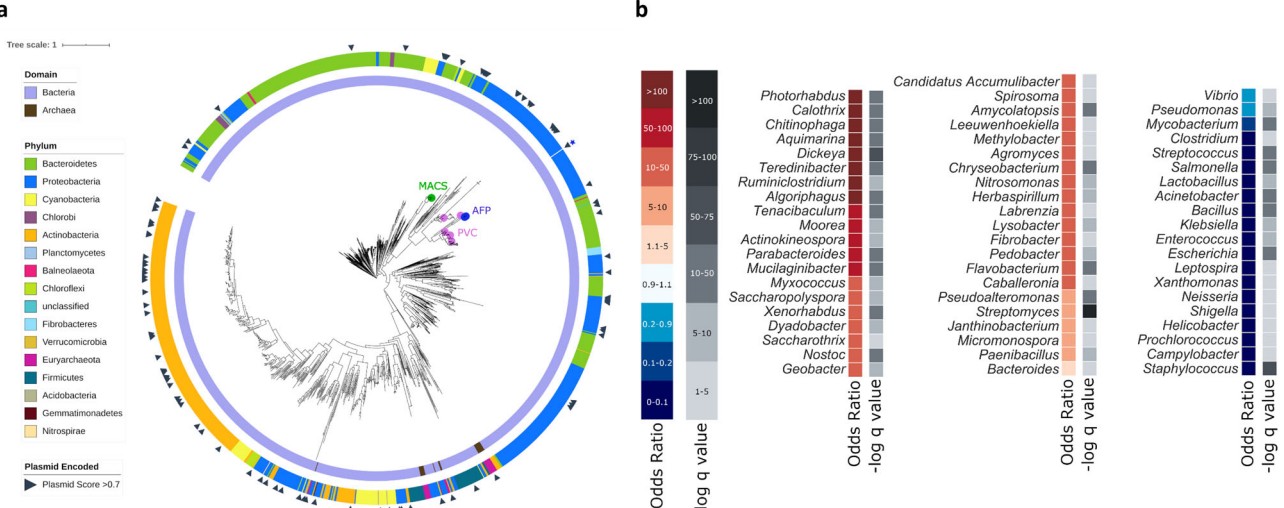

**Fig. 1 Taxonomic Distribution of eCIS-encoding microbes. a** A phylogenetic tree of eCIS across the microbial world. eCIS core genes Afp8 and Afp11 from each operon were concatenated, aligned, and used to construct the phylogenetic tree. The Domain and Phylum corresponding to each leaf are indicated in the inner and outer rings, respectively. Scaffolds encoding eCIS that have been predicted to be plasmids using Deeplasmid were marked with black triangles. Previously experimentally investigated eCIS are marked on their respective leaves (2 o'clock). Within the tree MACS, AFP, and PVC are abbreviations for Metamorphosis-associated Contractile Structures, Antifeeding Prophage, and *Photorhabdus* Virulence Cassettes. **b** eCIS distribution in different genera. We calculated the eCIS distribution across genera using a Fisher exact test. The Odds Ratio represents the enrichment or depletion magnitude, with hotter colors representing enrichment, and colder colors representing depletion. Calculated *p* values were corrected for multiple testing using FDR to yield minus log10 *q* values, shown in shades of gray. Only selected Genera are shown. Source data are provided in Supplementary Data 1–2,5–6.

are mostly constrained to particular taxa (Fig. 1b, Supplementary Data 6). Strikingly, we found that it is present in 100% (18/18) of *Photorhabdus* genomes in our dataset (Fisher exact test, odds ratio = infinity, *q* value = $2.97e^{-28}$), 89% of sequenced *Chitinophaga* (odds ratio = 276, *q* value = $1.69e^{-35}$), 86% of sequenced *Dickeya* (odds ratio = 211, *q* value = $3.78e^{-18}$), and 69% of sequenced *Algoriphagus* (odds ratio = 73, *q* value = $1.99e^{-24}$). These genera are known as environmental microbes; *Photorhabdus* is a commensal of entomopathogenic nematodes[20], *Chitinophaga* is a soil microbe and a fungal endosymbiont[21], *Dickeya* is a plant and pea aphid pathogen[22,23], and *Algoriphagus* is an aquatic or terrestrial microbe[24–28]. In contrast, eCIS is strongly depleted from the most cultured and sequenced genera of Gram-positive and negative human pathogens, including *Staphylococcus*, *Escherichia*, *Salmonella*, *Streptococcus*, *Acinetobacter*, and *Klebsiella*. Strikingly, within these genera, for which our repository had 18,355 genomes, eCIS was totally absent (odds ratio = 0, *q* value ≤ 3.86E-16 for each one of these genera), suggesting a very potent purifying selection acting against eCIS integration into these microbial genomes, despite the eCIS operons' tendency for extensive lateral transfer and its presence in other host-associated systems. Interestingly, 146 genomes, mostly from *Photorhabdus*, *Dickeya*, *Actinokineospora*, *Streptomyces*, *Algoriphagus*, *Chitinophaga*, *Flavobacterium*, and *Calothrix* genera, were found to contain more than one eCIS operon, ranging from 2 to 5 copies per genomes (Supplementary Data 7).

**eCIS presence is highly correlated with specific ecosystems, microbial lifestyles, and microbial hosts**. Given the strong eCIS taxonomic bias we identified, we were curious to know if we could further associate eCIS with specific ecological features. To this end, we retrieved metadata available for all sequenced genomes in our repository (Methods). These traits include the microbial isolation site, ecosystem and habitat, microbial lifestyle and physiology, and the organisms hosting the microbes (Supplementary Data 8). We calculated the correlation of eCIS

presence with certain microbial traits to identify significant enrichment and depletion patterns. This was done using a naïve enrichment test (Fisher exact test) together with a phylogeny-aware test, Scoary[29], which is used to correct for the phylogenetic bias of the isolate genomes. Using this test we quantify to what extent the eCIS presence in a genome correlates with a certain trait, independently of the microbial phylogeny (Fig. 2, Supplementary Fig. 8). Notably, eCIS is positively correlated with terrestrial and aquatic environments, such as soil, sediments, lakes, and coasts, but is depleted from food production venues. In terms of microbial lifestyle and physiology, eCISs are enriched in environmental microbes, mostly symbiotic, and are depleted from pathogens (the vast majority of which were isolated from humans). eCISs are enriched in aerobic microbes that dwell in mild and cold temperatures. In general, the eCIS-encoding microbes tend to associate with terrestrial hosts including insects, nematodes, annelids, protists, fungi, and plants, and in aquatic hosts such as fish, sponges, and molluscs. Intriguingly, we detected a strong depletion from bacteria that were isolated from birds and mammals, including humans. We did find some eCIS isolated from bacteria associated with humans, but sparse and statistically depleted (Supplementary Fig. 8). Looking closer we also see that the operon is depleted from all tissues in which the human microbiome is abundant: oral and digestive systems, skin, and the urogenital tract. However, we detected a mild eCIS enrichment in the human gut commensal *Bacteroides* (Fig. 1b) and *Parabacteroidetes* genera. *Bacteroides* was recently reported by the Shikuma group as being eCIS-rich[30].

We also see that eCIS is clearly associated with larger bacterial genomes in five bacterial phyla (Supplementary Fig. 9), although small genome endosymbionts are found to contain eCIS as well, for example, the *Candidatus Regiella insecticola LSR1*, which harbours an eCIS even though its genome size is ~2 Mbps and it contains <2000 genes[31].

Given these findings, we conclude that eCIS is an environmental microbial secretion system. Moreover, to the best of our

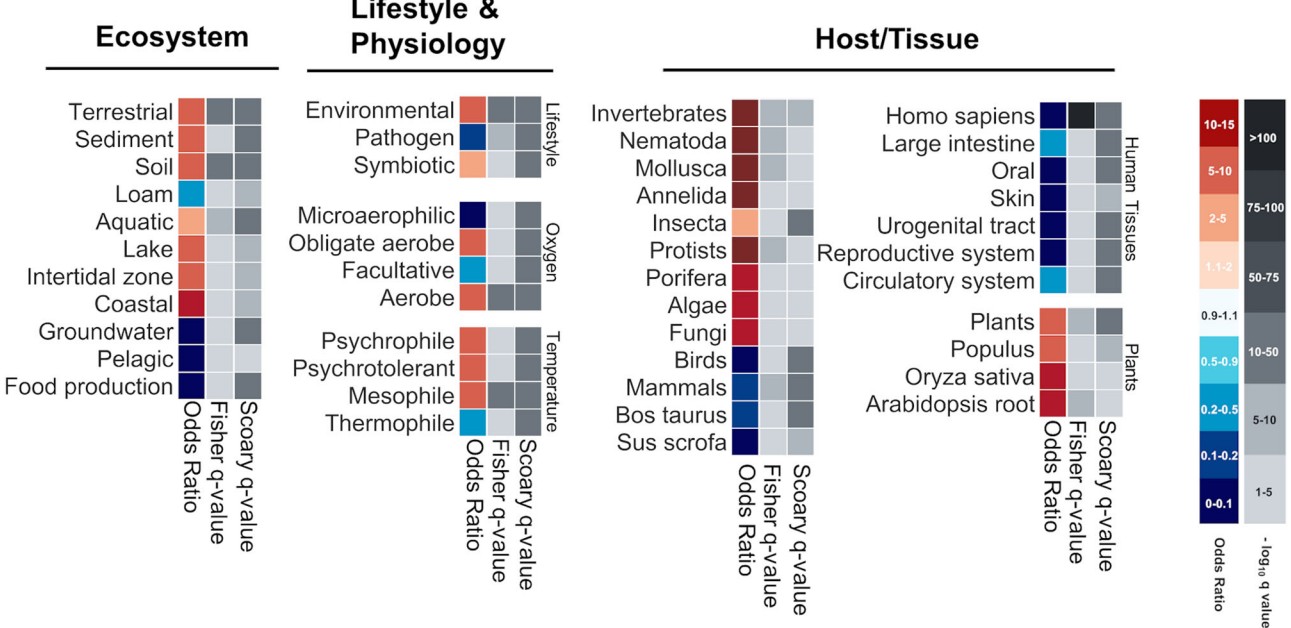

**Fig. 2 eCIS-encoding microbes' lifestyle and isolation.** A Fisher exact test combined with a modified version of Scoary was used to perform a phylogeny-aware analysis of eCIS-encoding microbes' metadata. The Odds Ratio represents the enrichment or depletion magnitude, with hotter colors representing enrichment, and colder colors representing depletion. The negative log10 of the q-values, shown in shades of gray, are corrected for multiple hypothesis testing. One q-value corresponds to the statistical significance of a two-sided Fisher exact test, and the other represents the same for the Scoary pairwise comparison test. Source Data are provided in Supplementary Data 8.

knowledge, this is the first secretion system that exhibits such a strong genomic enrichment in environmental microbes and a depletion from mammalian and avian microbiomes. Therefore, we hypothesize that this environmental association points to the eCIS biological function and target organism specificity.

**eCIS tail fibers provide information about target specificity and suggest targeting of bacteria.** All previously studied eCIS target invertebrate cells. One of the proteins that likely mediates specific eCIS binding to target cells prior to effector injection is the tail fiber protein that is attached to the particle's baseplate[32]. This protein is encoded by the *Afp13* gene and has homologues across the different eCIS operons. The Afp13 from *Serratia* spp. was shown to share amino acid sequence similarity with the tail fiber protein of adenovirus, a eukaryotic-targeting virus[6,11]. We sought to classify the Afp13 homologues in our dataset according to the fiber protein they are most similar to, whether from eukaryote-targeting viruses or from bacteria-targeting viruses (phages) to assign potential eCIS target cells. To do so we compiled a database of all proteins from 192,410 eukaryote-targeting viruses and 12,508 tailed phages and used protein BLAST with 629 Afp13 protein sequences as the query (Supplementary Data 9, Supplementary Figs. 10–14, Methods). In many eCIS operons we could not detect *Afp13* homologues. We identified Afp13 that have strong matches to eukaryote-targeting viruses, including fiber proteins from Equine Adenovirus 1, Bat mastadenovirus B, and Rhesus adenovirus 60. Interestingly, hits were also detected against the algae-infecting Organic Lake Phycodnavirus, as well as against amoeba-infecting Yasminevirus (Supplementary Fig. 10, Supplementary Data 9), suggesting new eCIS eukaryotic targets beyond invertebrates. Our data showed that 76 Afp13 sequences matched eukaryotic-targeting viruses better than they matched phages. However, four Afp13 sequences aligned best with phage fibers (Supplementary Figs. 10–14, Supplementary Data 9). We therefore propose that in these cases, the eCIS particles likely target bacteria. This notion is supported by the fact that highly-

related contractile tail particles, R-type pyocins, have analogous tail fibers that indeed specify host range by binding to specific strains of bacteria[33]. We did not detect a separate clustering of the putative phage-binding and virus-binding Afp13 (Supplementary Fig. 11).

These aforementioned cases showed that some tail fibers had clear homology that could be predictive of target. However, overall, most of the 629 tail fiber genes did not have any hit to either the phage nor Eukaryotic-targeting virus. Those that did have a hit mostly looked like both phage and eukaryotic-targeting viruses in equal measure (Supplementary Figs. 11–14, Supplementary Data 9, Methods). Notably, only 18% of the Afp13 protein sequences tested share a clear sequence similarity to a known tail fiber from phages or from eukaryotic-targeting viruses (rather than to both groups) (Supplementary Figs. 11–14, Supplementary Data 9). Therefore, in the majority of the cases, it is challenging to predict the kingdom to which the eCIS target organism belongs solely based on tail fiber protein sequence.

**Characterization of the eCIS known and new core and shell protein domains.** Hundreds of eCIS operons were previously defined[1,17] but a systematic analysis of the genes that compose the eCIS operons has not been carried out yet. To comprehensively characterize the eCIS repertoire of genes, we compiled the list of Protein Family (pfam) domains across the 1425 eCIS operons in our database (Methods). We then performed an enrichment test to identify protein domains that are statistically enriched in eCIS operons in comparison to all other genes in our 64,756 genome dataset (Fig. 3a, Supplementary Data 9). As expected, most of the known core eCIS protein domains are highly enriched and are present in nearly all eCIS loci (Fig. 3a, b). These include the domains encoding eCIS tail tube, spike complex, sheath, baseplate, tail terminator protein, and ATP supply (Fig. 3a). To identify yet unknown core domains we searched for protein domains that are present in eCIS from at least ten microbial phyla. By doing so we identified a highly conserved eCIS core domain of unknown

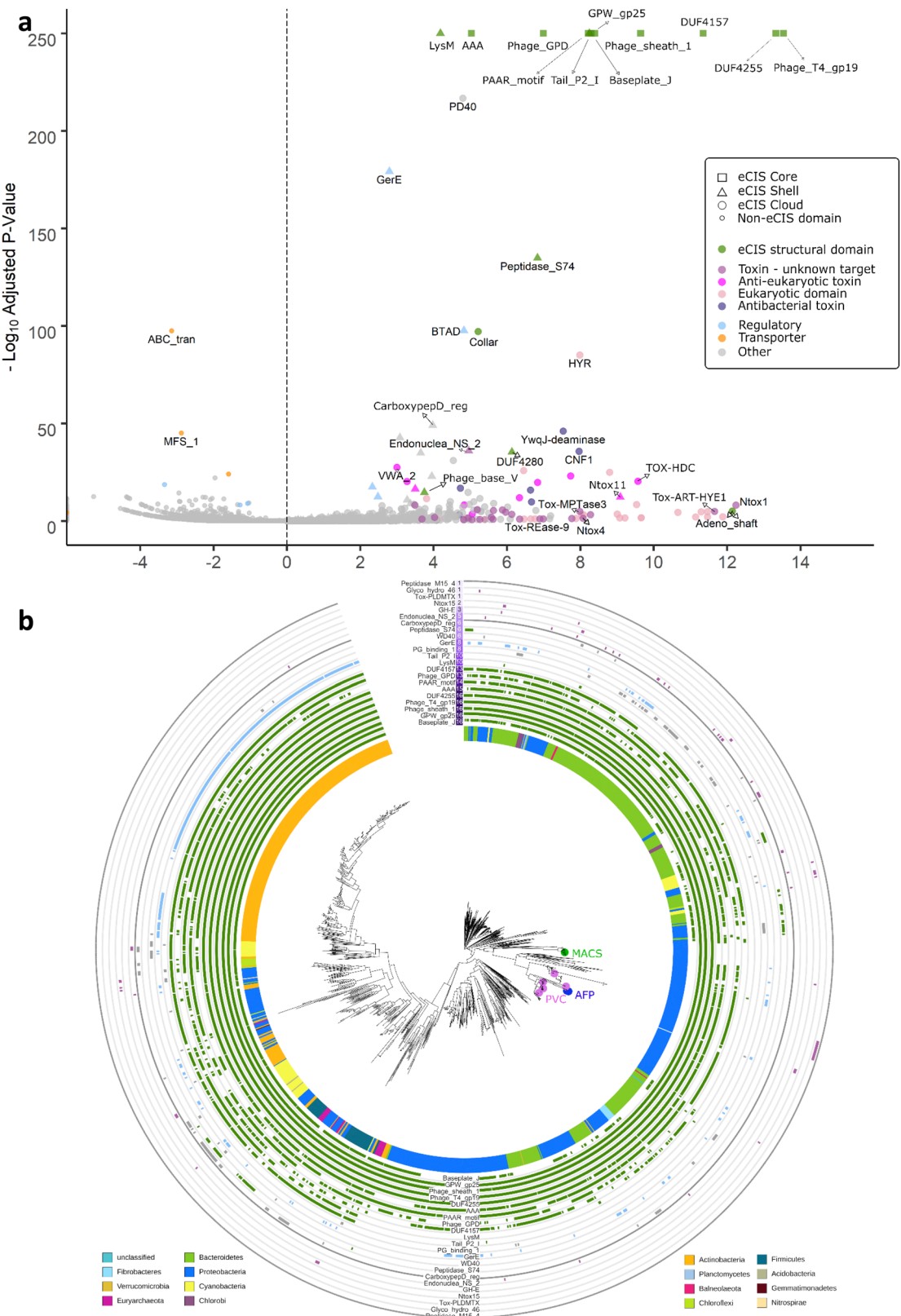

function 4157 (DUF4157). DUF4157 is found in 13 phyla and has high eCIS specificity (odd ratio = 2614, $q$ value = 0). It contains the Zn-binding motif HExxH, which characterizes many metalloproteases[34–37]. This domain, previously termed PVC-metallopeptidase, was identified as a marker of eCIS based on limited genome analyses[38,39] but its role has never been studied in a broad and functional context. By analyzing the eCIS operons and proteins we identified that DUF4157 is strongly associated with a high number of putative eCIS toxins, either as a gene that is adjacent to toxins or in the form of a multidomain protein, in which DUF4157 is found at the N-terminus and the toxin domain is found at the C-terminus (Supplementary Fig. 15). Interestingly,

**Fig. 3 Protein Family (Pfam) domains that are enriched in eCIS. a** Volcano plot of pfam domain enrichment in eCIS operons. Fisher Exact test corrected with Benjamini-Hochberg procedure. X axis is Log2 of odds ratio, Y axis is negative log10 of corrected P-value. The shape of each point is represented by the number of Phyla containing the Pfam in eCIS operon; >10 is defined "Core", 4–10 is "Shell", <4 is "Cloud". The color of each point represents a functional context of the Pfam domain. The known eCIS core domains encode the tail tube (Phage_T4_gp19), spike complex (PAAR_motif, Phage_GPD), sheath (Phage_sheath_1), baseplate (GPW_gp25, Baseplate_J), tail terminator protein (DUF4255), and ATP supply (AAA). DUF4157 is a new eCIS core domain. **b** Phylogenetic distribution of eCIS-associated pfams. At the 12 o'clock and 6 o'clock positions of the tree are labels for select eCIS-associated pfams. Next to each pfam domain name at 12 o'clock is a heatmap quantifying in how many phyla a given pfam is found in, along with a ring corresponding to each label (surrounding the circumference of the tree) with colors corresponding to (**a**). Within the tree MACS, AFP, and PVC are abbreviations for Metamorphosis-associated Contractile Structures, Antifeeding Prophage, and *Photorhabdus* Virulence Cassettes. Source Data are provided in Supplementary Data 10.

in many cases there is a very long linker sequence between DUF4157 and the various toxin domains, occasionally more than 1000 amino acids long (Supplementary Fig. 15). The role of the DUF4157 domain in eCIS remains to be studied. We hypothesize that it is related to toxin loading, release, maturation, or trafficking inside the target cell.

Next, we defined 'shell' domains as those that are present in eCIS from 4 to 10 microbial phyla and likely produce specific eCIS subtypes or are responsible for regulation (Fig. 3, Supplementary Data 10). These shell domains include, for example, LysM (PF01476), Tail_P2_I (PF09684), and the bacterial transcriptional activator domain (BTAD, PF03704). An example of a new eCIS shell domain is the GerE transcriptional regulator (PF00196), which is a LuxR-type DNA-binding domain. We find GerE located next to 25.8% of the eCIS loci, thereby defining it as the first conserved eCIS transcriptional regulator (Fig. 3a, b). In 9% of the eCIS operons containing the GerE transcriptional regulator, a histidine kinase gene was found upstream to the GerE genes (Supplementary Fig. 16). These two genes might constitute a two-component system regulating the eCIS operon. Another shell domain is the Peptidase S74 (PF13884). The Peptidase S74 is a known domain of the intramolecular chaperone of the bacteriophage T5 tail fiber[40] (Supplementary Fig. 17). eCIS operons containing Peptidase S74 genes did not have annotated tail fibers (*Afp13*), suggesting they might have similar functions or that peptidase S74 functions with a yet unknown Afp13 analog (Supplementary Fig. 14). In addition, we looked for a unique pfam domain distribution between Archaea, Gram-positive and Gram-negative bacteria (Supplementary Fig. 18). Although many unique pfam domains can be found in each of the groups, none of them is highly specific to and characteristic of a single taxonomic group. We hypothesize that the unique pfams are a consequence of variability in lower taxonomic levels.

**A large collection of toxin domains is encoded within eCIS operon genes.** We identified numerous toxin domains located next to eCIS core components. Intriguingly, unlike the core and shell domains, most putative toxins are specific to certain eCIS loci (Fig. 3b), suggesting rapid evolution and diversification of these toxins as part of the arms race against different target cells. Some of these domains are likely antibacterial toxins supporting the idea that some eCIS are released to target bacteria, such pesticin (PF16754) which degrades peptidoglycan[41] encoded within one of the eCIS loci of *Pseudoalteromonas luteoviolacea*. Other toxins, as expected, seem to target eukaryotes, e.g.,: cycle inhibiting factor (PF16374) that arrest cell cycle[42], and Von Willebrand factor type A domain, which is a glycoprotein found in blood plasma (PF00092)[43]. In addition, we predict that other eCIS domains serve as eCIS anti-eukaryotic toxin effectors as they are mostly found in eukaryotic proteins (Supplementary Fig. 19), such as hemopexin that is found in plasma proteins (PF00045)[44], Annexin (PF00191), and the animal-specific prominin domain

(PF05478)[45]. Eukaryotic protein domains are common in effectors translocated into eukaryotic cells by various secretion systems[46–49]. Overall, we identified at least 71 protein domains that are likely toxins and are found next to eCIS core genes and compiled a list of 496 genes that contain at least one of these putatively toxic domains (Supplementary Data 10–11).

**Experimental validation of 12 new eCIS-associated toxins that target bacteria.** Next, we decided to test our computational toxin gene predictions. We tested the killing of *E. coli* as we predict that certain eCIS particles inject toxins into bacteria. To identify new toxins we first clustered all eCIS proteins based on sequence similarity and annotated core genes (Methods, Supplementary Data 12). We selected 21 candidate eCIS toxins (Supplementary Table 2) based on several features, such as small size, encoding domains (e.g., proteins with DUF4157), and location at the 3' end of their respective operon[4,5,11]. We synthesized and cloned genes of interest under an inducible T7 promoter in pET vectors, transformed these into *E. coli*, and induced expression using IPTG. We used a known antibacterial toxin, Tse2[50], as a positive control, and two negative controls: a monomeric enhanced GFP gene[51] and an eCIS phage tail protein that is not expected to be toxic (locus tag Ga0070536_101452). Following serial dilutions of the colonies we could quantify the expressed protein toxicity levels. Out of the 20 genes tested, 12 were found to be toxic to *E. coli* (Fig. 4a, Supplementary Figs. 20–21, Table 1, Supplementary Table 2). We designate these as eCIS-associated toxins ("EATs") 1–12. One candidate EAT gene failed cloning, likely due to high toxicity even without expression induction (Supplementary Table 2). Based on sensitive sequence similarity search we identified that the EATs likely encode various molecular functions, such as ADP ribosylation, peptidase, NAD + phosphorylase, endonuclease, peptidoglycan hydrolysis, and deaminase (Fig. 4b, Table 1, Methods). Most importantly, these are the first EATs demonstrating antibacterial activity. Seven EATs include DUF4157 domains as part of the protein or in a nearby gene (Table 1). We tested for protein expression of five of the predicted EATs that were not found to be toxic. We saw that three of these were expressed and are therefore indeed non-toxic to *E. coli* and two were not detected and therefore their heterologous expression may have been impaired (Supplementary Fig. 22, Supplementary Table 2). EAT10 was predicted to function as a peptidoglycan hydrolase. Therefore, we hypothesized that it should be more toxic to *E. coli* upon expression in the periplasm than in the cytoplasm. We cloned EAT10 into pBAD24 vector with or without the twin-arginine leader motif that translocates proteins into the periplasm and indeed observed a $10^4$ higher toxicity in the periplasm than in the cytoplasm (Supplementary Fig. 23). This result provides yet another support for potential EAT antibacterial activity since the periplasm is a bacteria-specific compartment.

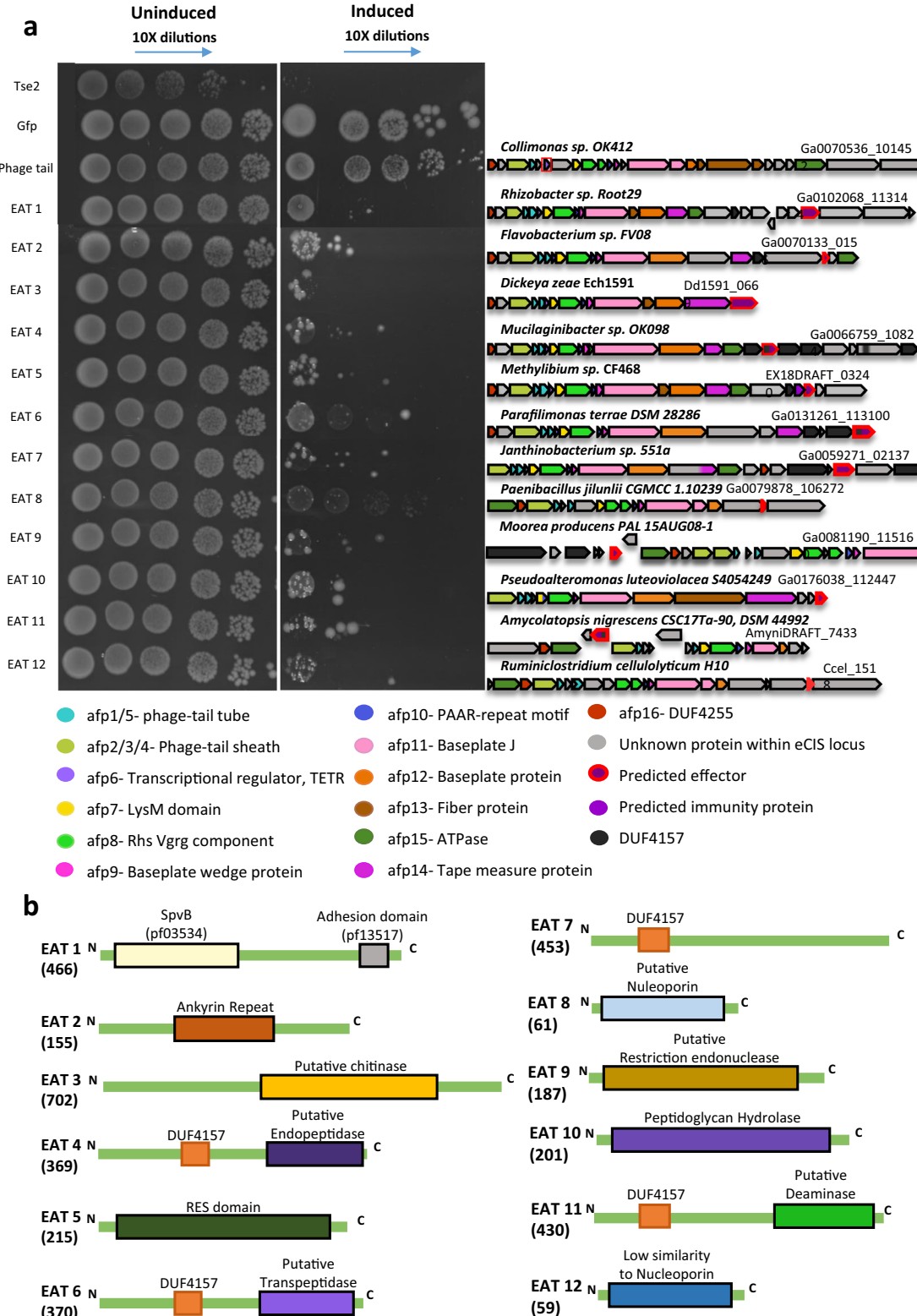

**Fig. 4 EATs are toxic to *E. coli*. a** EATs were cloned into pET28/29, transformed into *E. coli*, and were serially diluted and plated onto agar in induction conditions using 500 μM IPTG (right panel) or non-inducing conditions (left panel). tse2, a known Type VI Secretion System effector, was used as a positive control. Two negative controls used were meGFP, and an eCIS gene (phage tail). On the right side of each experiment is a cartoon representing the operon of the eCIS which the EAT was cloned from (purple gene with a red border); below is a legend explaining each component's color. Image shows representative results of one experiment out of two experiments with similar results. The full results are provided as a Source Data file. **b** Schematic representation of EAT protein and the domains within them. Size in amino acids is indicated in parentheses.

**Table 1 New eCIS-associated toxins (EATs) experimentally validated in the current study.**

| Toxin name | Gene ID (IMG gene id) | Organism (taxon) | Activity | Predicted molecular function | DUF4157 gene/protein association | Tail fiber similarity |
|---|---|---|---|---|---|---|
| EAT1 | Ga0102068_113144 (2644141250) | *Rhizobacter* sp. Root29 (Betaproteobacteria) | Antibact., anti-euk. | ADP-ribosyl transferase (SpvB) and adhesion (VCBS) | Absent | No clear hit toward either phage or virus (gene 2644141239) |
| EAT2 | Ga0070133_0156 (2616189889) | *Flavobacterium* sp. FV08 (Bacteroidetes) | Antibact., did not kill yeast. | Unknown. Encodes ankyrin repeats. | One and two genes upstream to EAT2 | No afp13 in operon |
| EAT3 | Dd1591_0669 (644851057) | *Dickeya zeae* Ech1591 (Gammaproteo.) | Antibact., anti-euk. | Chitinase? | Absent | No clear hit toward either phage or virus (gene 644851060) |
| EAT4 | Ga0066759_10824 (2609591113) | *Mucilaginibacter* sp. OK098 (Bacteroidetes) | Antibact., did not kill yeast. | peptidyl-Lys metalloendopeptidase | Within protein center | No afp13 in operon |
| EAT5 | EX18DRAFT_03240 (2587734256) | *Methylibium* sp. CF468 (Betaproteobacteria) | Antibact. | NAD + Phosphorylase | Two genes upstream to EAT5. | Phage-like (gene 2587734249) |
| EAT6 | Ga0131261_113100 (2695001213) | *Parafilimonas terrae* DSM 28286 (Bacteroidetes) | Antibact., did not kill yeast. | L,D-transpeptidase | Within protein | No afp13 in operon |
| EAT7 | Ga0059271_02137 (2602024578) | *Janthinobacterium* sp. 551a (Betaproteobacteria) | Antibact. | Unknown | Within protein (N terminus) | No afp13 in operon |
| EAT8 | Ga0079878_106272 (2668039404) | *Paenibacillus jilunlii* CGMCC 1.10239 (Firmicutes) | Antibact., did not kill yeast. | Nucleoporin? | Absent | No afp13 in operon |
| EAT9 | Ga0081190_115160 (2631153083) | *Moorea producens* PAL 15AUG08-1 (Cyanobacteria) | Antibact., did not kill yeast. | Restriction endonuclease | Three genes upstream to EAT9. | No afp13 in operon |
| EAT10 | Ga0176038_112447 (2720476818) | *Pseudoalteromonas luteoviolacea* S4054249 (Gammaproteo.) | Antibact. | Peptidoglycan hydrolysis | Absent | No clear hit toward either phage or virus (gene 2720476822) |
| EAT11 | AmyniDRAFT_7433 (2515141031) | *Amycolatopsis nigrescens* CSC17Ta-90, DSM 44992 (Actinobacteria) | Antibact., anti-euk. | Deaminase | Within protein (N terminus) | No afp13 in operon |
| EAT12 | Ccel_1518 (643608442) | *Ruminiclostridium cellulolyticum* H10 (Firmicutes) | Antibact. | Nucleoporin? | Absent | No afp13 in operon |
| EAT13 | Ga0111076_123119 (2653242303) | *Candidatus Udaeobacter copiosus* (Verrucomicrobia) | Anti-euk. | Actin cross-linking | Flanked by genes with DUF4157 domain | No afp13 in operon |

Antibact.—toxins killing the bacterium *E. coli*. Anti-euk.—toxins killing *the yeast S. cerevisiae*.

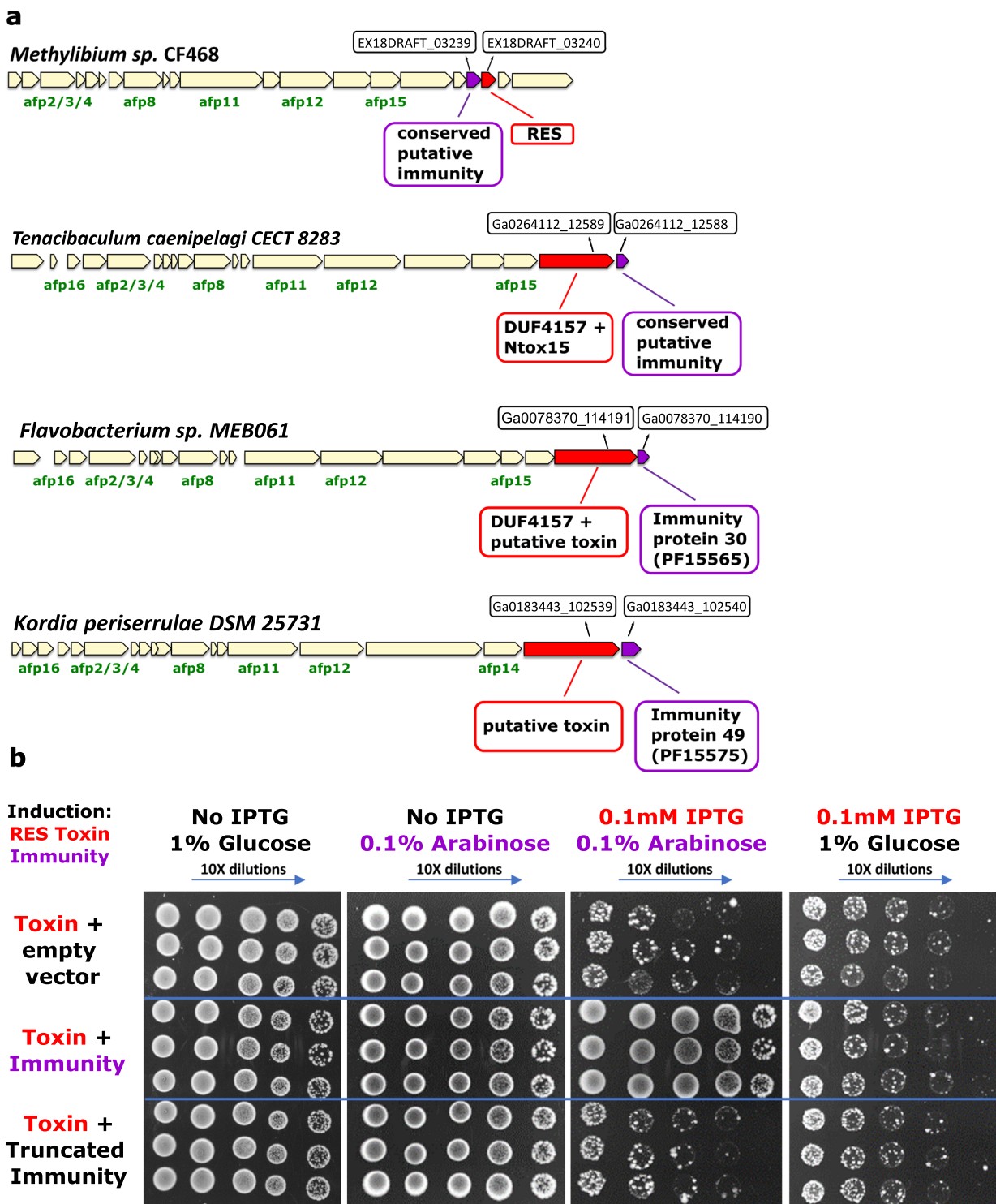

**Fig. 5 Toxin-immunity gene pairs are genetically associated with eCIS. a** Pairs of known antibacterial toxins found in eCIS. eCIS operons containing known or putative antibacterial toxin or immunity domains. **b** XRE-like antitoxin protects from RES-like toxin (EAT5). Induction by IPTG leads to expression of RES-like toxin (IMG locus tag EX18DRAFT_03240). Glucose represses XRE-like antitoxin (IMG locus tag EX18DRAFT_03239) or its truncated version while Arabinose induces expression of XRE-like antitoxin (or its truncated version). Each section (separated by blue lines) represents three biological repeats.

**Several eCIS-associated toxins are genetically linked to anti-toxins.** Although many of the tested EATs above puzzlingly do not have putative immunity genes associated with them, we identified some cases of toxin-immunity gene pairs next to eCIS operons supporting the antibacterial role of these eCIS loci as they require immunity genes (antitoxins) to protect from

self-intoxication (Fig. 5a). In one case, the toxin, EAT5, resembles RES (Fig. 4b), an NADase that is accompanied by an antitoxin called Xre[52,53]. We identified a gene, EX18DRAFT_03239, encoding an unknown protein located upstream of the toxin. We predicted that it serves as a new cognate immunity gene for the RES family, as the gene pair organization is conserved in

phylogenetically diverse bacterial genomes (Supplementary Fig. 24). Expression of the RES-like EAT5 toxin was indeed toxic to *E. coli* while the putative immunity gene was not (Fig. 5b). Co-expression of the toxin-immunity pair was sufficient to rescue bacteria from toxicity (Fig. 5b). Finally, we showed that an intact immunity protein is required for the rescue phenotype, since deletion of 17 amino acids which encompass the C terminal alpha helix domain from the antitoxin abolished the immunity function (Fig. 5b). Interestingly, the tail fiber protein (encoded by gene EX18DRAFT_03233) of the EAT5-associated eCIS resembles a protein from *Thermus* phage YS40_Isch (Table 1; gene 2587734249 at Supplementary Data 3), providing additional evidence that this eCIS targets bacteria and is therefore accompanied by an immunity protein.

Putting the data together, we present here the first 12 antibacterial EATs and the first EAT-immunity gene pairs.

**Experimental validation of new four eCIS toxins that target eukaryotic cells**. We were also interested in discovery of eCIS anti-eukaryotic toxins. We predicted new EATs that target eukaryotes based on their presence in the 3′ end of the eCIS operon, presence of eukaryotic domains within the encoded proteins, predicted enzymatic activity on eukaryote-specific molecules (such as actin), and similarity to known virulence factors (Supplementary Table 2). We cloned 14 genes into pESC-leu galactose-inducible vectors and transformed them into *S. cerevisiae* cells, serving as models for eukaryotic cells (Methods). Upon induction, we identified four EATs that efficiently killed yeast (Fig. 6, Supplementary Fig. 25). Three EATs, EAT1, EAT3, and EAT11, also killed *E. coli*. EAT13 gene was tested only in yeast. These are predicted to act as an ADP-ribosyl transferase, a chitinase, a deaminase, and an actin cross-linker, respectively (Table 1, Figs. 4b, 6c). We think that yeast is probably not the native target for many other anti-eukaryotic EATs and therefore, it is likely that some of the toxins we predicted may be active when expressed in different organisms found in the native habitat of eCIS-encoding microbes. Importantly, EATs 2,4,6,8, and 9 were tested in both *E. coli* and *S. cerevisiae*, and we only found toxicity in *E. coli*, demonstrating specific antibacterial activity.

**Development of a comprehensive eCIS database**. Finally, we developed a repository called eCIStem that contains 1425 operons from 1249 bacterial and archaeal genomes. The repository is freely available for academic users at: http://ecistem.pythonanywhere.com. eCIStem provides visualization of all eCIS operons and includes gene annotation, an eCIS operon search option based on taxonomy, pfam domains, and gene clusters, and links to additional gene and protein data from the IMG database[16]. eCIStem also contains ecological and lifestyle metadata of the eCIS-encoding microbe. This valuable information can help researchers to study eCIS with regard to its specific ecological functions and to deduce the eCIS target organism. Supplementary Figs. 26 and 27 present some of the information held in eCIStem. eCIStem is a significant expansion of the recently developed dbeCIS which includes 631 operons[17]. We also expanded dbeCIS functionality by providing protein domain data for all eCIS genes, eCIS-encoding organism metadata, and a full database download option which will facilitate future bioinformatic analysis of our dataset by the entire research community.

## Discussion

In this study we use bioinformatic, phylogenetic, and molecular methods in order to thoroughly characterize the eCIS system across the microbial world. Based on our results we provide an improved model for eCIS function that enriches our understanding of eCIS ecological and taxonomic distribution within microbes, potential eCIS regulation and cell attachment mechanisms. Moreover, we expanded the known repertoire of eCIS targets. We show that, most likely, eCISs not only target eukaryotic cells, as previously reported, but also bacteria (Fig. 7).

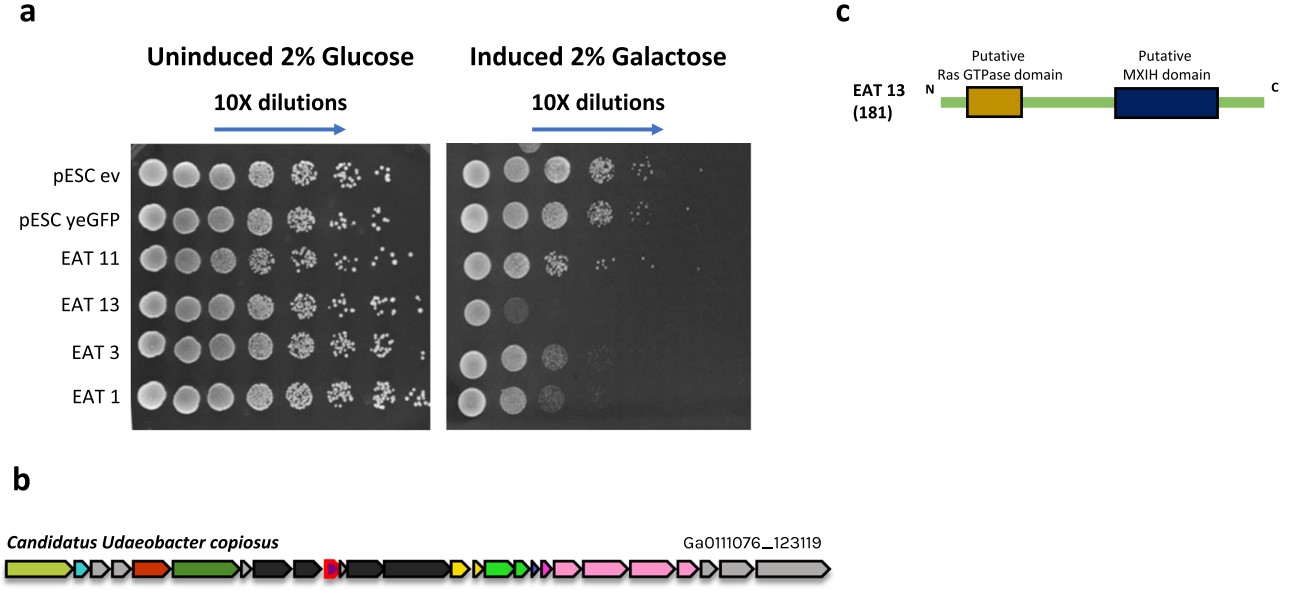

**Fig. 6 EATs are toxic to *S. cerevisiae*. a** EATs were cloned into pESC -leu Galactose inducible plasmids that were then transformed into *Saccharomyces cerevisiae* BY4742 strain. Overnight cultures of the strains harboring the vectors of interest were grown in SD -leu media. The cultures OD was normalized and then washed once with water and split into two: one part was grown overnight in repressive conditions (SD -leu + 2% glucose) and the other part was grown in inductive conditions (SD -leu + 2% galactose). Dilutions were spotted on SD -leu plates containing glucose or galactose and the plates were incubated two nights at 30 °C. Negative controls: empty vector (ev) and non-toxin (yeGFP gene). Three biological replicates of this experiment are presented in Supplementary Fig. 25. **b** Operon containing EAT13 toxin (outlined in red border). Gene color code matches that of genes in Fig. 4a. **c** EAT13 protein domains with predicted functions.

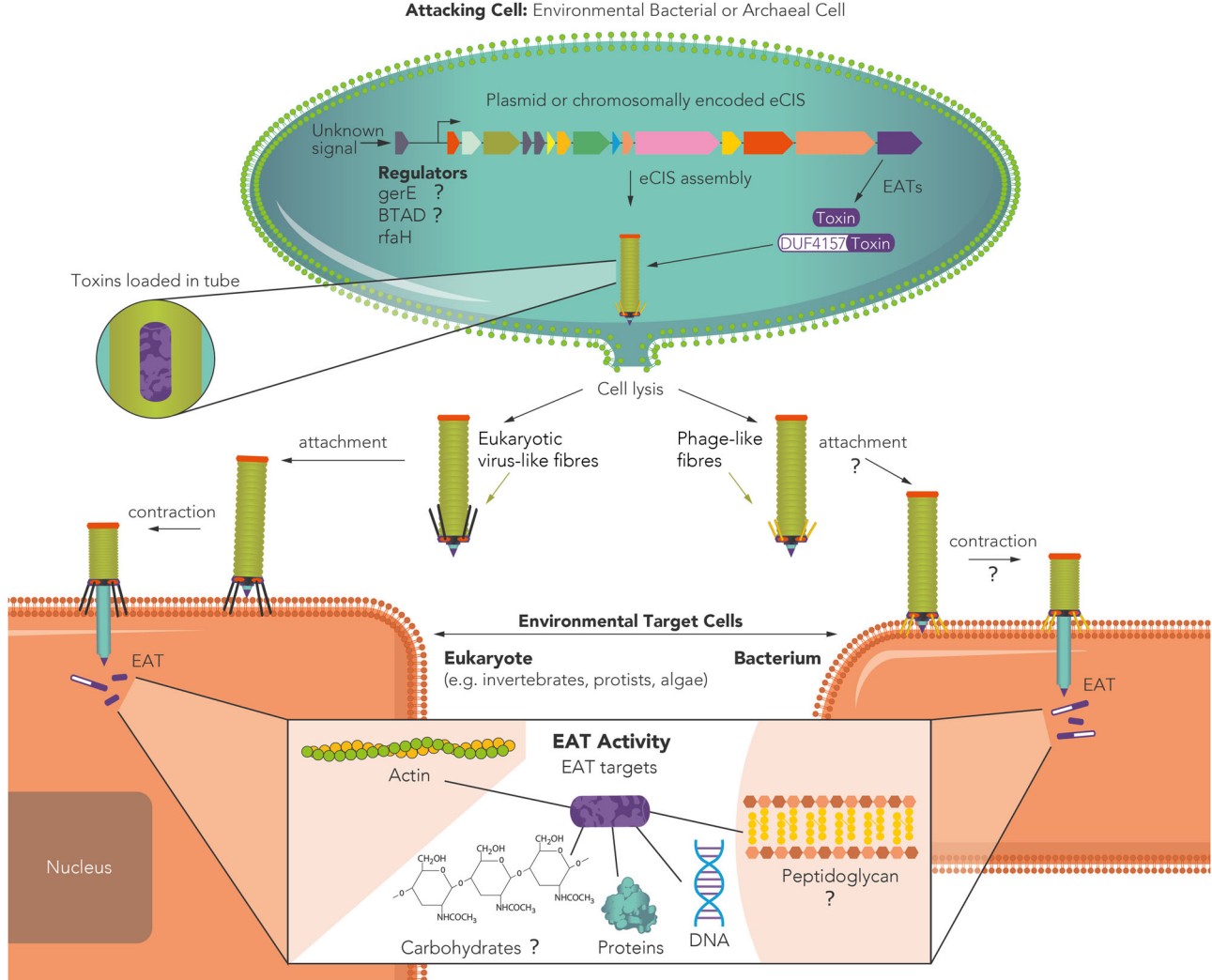

**Fig. 7 A revised model for eCIS function and ecology.** eCIS is encoded by operons that are enriched in environmental microbes. These operons are regulated by adjacent genes such as RfaH[78] or genes that carry GerE or BTAD domains. The regulators are occasionally activated by kinases and are activated/repressed by yet unknown signals. eCIS-associated toxins (EATs) tend to be encoded in the 3′ end of the operons. EATs often carry DUF4157 domains in their N-terminus or found adjacent to genes with this domain. The eCIS particle is assembled with the toxins inside the tube[15] and is released following cell lysis[13]. The particles carry tail fibers that mediate attachment to environmental eukaryotic or prokaryotic target cells. The tail fibers attach, contract, and release the EATs inside the target cells, mostly leading to its death or growth arrest. The EATs function intracellularly in diverse ways, enzymatically acting on different cellular components that were identified in the current and in previous works. Question marks denote aspects of eCIS function that are predicted based on our results yet have not been experimentally confirmed.

Our extensive analysis uncovered conserved eCIS core and shell components that encode the eCIS structure and regulation and numerous EATs genetically associated with eCIS loci. The high number and variety of EATs suggest that eCIS loci can serve as a treasure trove for toxin gene discovery. Previous works uncovered seven EATs, all from Gammaproteobacteria class, whereas here we added 13 EATs from six different bacterial phyla. We examine the occurrence of eCIS operons within a large-scale genome set and provide a strong support for eCIS enrichment in environmental microbes from soil and aquatic ecosystems and hosts, and a depletion from mammalian and avian microbiomes; with a specific depletion from their pathogens.

We identified that some eCIS tail fibers can help predict whether eCIS target bacteria or eukaryotes. Interestingly we identified eCIS tail fibers that share high sequence similarity with similar proteins of *Yasminevirus* giant virus and *Phycodnavirus*, which infect amoebae[54] and algae[55], respectively (Supplementary Fig. 10). Based on this finding we propose that eCIS from these

bacteria could target similar hosts. The fact that most eCIS tail fibers had similar identity with both phage and eukaryotic-targeting viruses suggests they may have broad spectrum binding activity, and therefore broad targeting activity (Supplementary Figs. 10–14).

One intriguing mystery is related to the antibacterial EATs and their lack of immunity genes in most cases. Nearly all antibacterial toxins produced by bacteria are accompanied by adjacent immunity genes[39,56–60], which we could not bioinformatically identify next to most of the newly verified EAT genes. We do not know how these antibacterial EATs are being produced and loaded onto the eCIS system without killing the producer cell prior to eCIS release. One possibility is that these are produced right before the cell is lysed to release the eCIS, and therefore the cell does not require protection at this time, as it is already "fated" to die, and cellular activity shifts to releasing eCIS particles. It is possible that EATs even purposely contribute to lysis. Another option is related to the yet mysterious function of the DUF4157

domain which accompanies many of the EATs. This domain may confer some immunity to the eCIS producers through temporal or spatial toxin inactivation.

Shedding light on eCIS biological role in nature can reveal new means by which microbes affect the health and development of their hosts and shape neighboring eukaryotic and prokaryotic microbial communities.

## Methods

**Identification of eCIS operons within microbial genomes.** The starting database for identification of eCIS was 64,756 publicly available genomes from the Integrated Microbial Genomes database[16], which includes all protein-coding genes along with their pfam annotations, as per the IMG pipeline. Genomes were searched for the presence of genes annotated as containing pfam domains associated with eCIS (Supplementary Table 1), for example, pfam14065, i.e., DUF4255, a domain in eCIS cap proteins[3]. Genes encoding proteins with the aforementioned pfam annotations were grouped by physical linkage (encoded within 12,000 base pairs of one another). Only those groups of linked genes with at least four members, with at least three of them having different pfam annotations, were considered putative eCIS operons. Because of the evolutionary similarity between eCIS and other contractile machinery, we removed any putative operons with genes annotated with phage-related pfams[61] (Supplementary Table 1), such as capsid genes (e.g., pfam03864, i.e., Phage major capsid protein E), within 10,000 base pairs of any genes in our putative operons. Furthermore, we removed any putative operons with genes annotated as COGs[62,63] associated with T6SS (Supplementary Table 1), within 10,000 base pairs of any genes in our putative operons, e.g., COG3523 (TssM, a membrane complex protein[63] absent in eCIS and phage). We believe this is a very stringent filtration protocol that was also added to filtration of pyocins (see below). The boundaries of these putative operons were expanded by ten genes up- and downstream of the extrema of the putative operons to identify eCIS accessory (non-core) genes. Only protein coding sequences were kept, and noncoding RNA genes were ignored. Using previously published HMM profiles of eCIS core elements[17], and the hmmsearch tool (a part of the HMMER package, version 3.3[64]), the expanded putative operons were surveyed and their genes were annotated as core elements. Hmmsearch results were ranked by bitscore, and we only considered results whose bitscore was in the top half of all scores, i.e., we dropped the bottom 50% of hits, in order to minimize false-positive hits. Furthermore, genes annotated by IMG as having pfams that match core elements were also annotated as core genes. The core-annotated putative operons were filtered for those with at least ten core genes, and with at least one eCIS-specific core (AFP 12, 13, 14, or 16[17]), in order to remove any possible R-type pyocins from the dataset. We chose to take the core genes on the extreme termini, and choose four genes up- and downstream to arbitrarily define the boundaries of the operon. The eCIS has inherently varied boundaries because (1) we see that many times the 3′ end has toxins, which, according to our "cloud" gene analysis, are quite varied in terms of their domains, and therefore in their structure; (2) we see evidence of horizontal gene transfer, thereby creating varied genes surrounding the eCIS operon; (3) core genes sometimes have multiple copies. All these considerations makes it hard to simply use an integer number of genes before and after the first and last core gene respectively to define boundaries. Our algorithm tended to be stringent, i.e., tended to minimize false positives, at the risk of possible false negatives. Given the high evolutionary similarity between eCIS and other related systems, we had a high threshold. For examples, we removed any eCIS with phage domains encoded nearby, which could be there by chance, and we only took those with at least one eCIS-specific core, with a high hmmsearch threshold, as described above. Because we perform detailed analysis, we did not want to introduce false positives into our database, and thereby were stringent. The final database of eCIS operons contains 1249 genomes, with 1425 distinct operons.

**Phylogenetic eCIS gene tree construction.** Amino acid sequences for Afp11 and Afp8 homologs were retrieved from each eCIS operon. In the case of operons with more than one copy of either the Afp8 or Afp11 homologs, the genes with the larger sequence were chosen. In the minority of cases (15/1425) where one or both of the genes were missing, they were not included in the tree. The Afp11 and Afp8 sequences were separately aligned with clustal omega version 1.2.4[65] using standard settings. The alignments were trimmed using TrimAl version 1.3[66] using standard parameters, plus the "automated1" flag. The Afp11 and Afp8 alignments from corresponding operons were concatenated. The concatenated alignments were used as input for FastTree version 2.1.11 using standard parameters[67], and visualized and annotated using interactive Tree Of Life (iTOL version 5.7[68]).

**Systematic tail fiber (Afp13) sequence analysis.** 629 Afp13 amino acid sequences, as defined by genes labeled by hmmsearch as afp13 within the top 25th percentile of bitscores, were used as queries for two searches using BLASTP against the NCBI nr database using standard parameters[69]. We expanded the AFP13 dataset to be wider than that in the eCIStem, which took the top 50th percentile, in order to get a broader picture of this relatively sparse gene. In the first search, the nr database was filtered by taxon IDs corresponding to tailed bacteriophages (NCBI

taxid: 28883). In the second search, the nr database was filtered by taxon IDs corresponding to eukaryotic-targeting viruses, by manually removing prokaryotic viruses from taxid 10239 (see Supplementary Data 9 for full taxid list). The results from each search were filtered for those hits with $p$ value < 0.001, leaving 296 unique Afp13 hits. Each query Afp13 gene was checked for the hit with the top bitscore from the first and second searches to determine if a given gene better aligned with tailed phages, or with eukaryotic-targeting viruses (Supplementary Data 9). Those Afp13 with bitscore difference >15 were marked as having a preference for one dataset over another; within 15 was considered inconclusive.

**Detection of eCIS within plasmids using Deeplasmid.** We developed Deeplasmid[18] (https://sourceforge.net/p/deeplasmid/). The Deeplasmid learning model was trained on data elements consisting of the label (plasmid vs. chromosome) and two input words: a 300 bp contiguous subsequence sampled randomly from the full original contig sequence and a vector containing 16 features extracted from the full sequence. The features we used included GC content, homopolymer content, hits to known chromosomal and plasmid proteins and ORIs, gene count and coding content, amino acid polypeptide length, and contig length. One branch of the deep learning network, consisting of sequence processing networks (LSTMs), accepts a 300 bp nucleotide sequence, and compresses information into a vector of 40 features. The other branch is fully connected, accepts the feature vector, and produces a vector of 20 features. The outputs of these two parallel branches are concatenated and passed to another block of fully connected layers whose output is the final classifier score y. Scores above a threshold 0.55 are plasmids, under 0.45 are chromosomal, and middle values are ambiguous. The model was trained with a binary cross-entropy loss function and Adam optimizer. We performed supervised learning on a balanced set of known chromosome and plasmid scaffolds. The model was trained for 30 epochs, until it converged. Deeplasmid achieves an AUC-ROC of over 93% on a separate dataset of ~6000 sequences from IMG[16]. The AUC-ROC is a performance measurement for classification at various thresholds settings. AUC represents how well the model is capable of distinguishing between classes. The ROC curve is plotted as the true-positive rate (TPR) against the false-positive rate (FPR) where TPR is on $y$-axis and FPR is on the $x$-axis. A higher AUC means the model is better at predicting chromosomes as chromosomes and plasmids as plasmids. Comparison of Deeplasmid against PLASFlow[70] and cBar[71] on the IMG database shows Deeplasmid achieves a higher true-positive rate for predicted plasmid sequences of 94% (purity or precision) and a comparative recall of 77% for known plasmids predicted correctly.

We ran Deeplasmid on all scaffolds carrying eCIS. The score cutoff of $> = 0.7$ was chosen as it correctly identifies the pADAP plasmid that encodes the AFP. Other known plasmids carrying eCIS received Deeplasmid scores higher than 0.95 (Supplementary Data 5).

**eCIS association with genera.** The taxonomic makeup of the eCIS database was compared to the number of genera in publicly available genomes of Bacteria, Archaea, and plasmids of bacteria from IMG using a custom python script. A Fisher exact test was calculated for each genus, which returned an odds ratio and an associated $p$ value. Multiple hypothesis testing correction was performed using the Benjamini-Hochberg procedure. Those genera with a $q$ value of <0.001 and at least ten entries in the publicly available IMG database are displayed in Fig. 1.

**eCIS association with ecological and physiological features.** To correct for taxonomic bias that affects enrichment analysis, a population-aware enrichment analysis based on Scoary version 1.6.16[29] was used. Scoary accepts input of a tree to analyze Fisher exact test enrichments in the context of phylogeny. To create the tree, 16s sequences for each genome with eCIS was queried. In the case of more than one 16s gene per genome, one was chosen randomly. These sequences were clustered with clustal omega version 1.2.4[65], and FastTree[67] was used to make maximum likelihood tree (Supplementary Fig. 28). The inputs to Scoary were (1) the guide tree, (2) a presence-absence file, indicating if a genome has or lacks an eCIS operon, (3) metadata files that indicate for each genome if they possess a certain characteristic, e.g., whether it was isolated from soil. This program outputs an odds ratio, a $q$ value for the Fisher exact test that produced the odds ratio, and a $q$ value based on how many contrasting pairs (instances not from the same clade) support the enrichment, i.e., how many examples of the enrichment are independent of taxonomy.

**Protein domain (Pfam) enrichment analysis.** We collected the pfam annotations for the 64,756 proteins encoded by publicly available genomes from IMG. Based on our eCIS operon database, we marked all the pfam domains that were found in our eCIS database as "pfam domains within eCIS operons". Duplicated domains within a single gene (usually of repeat domains) were dropped, in order to avoid inflation of enrichment. We then counted all the occurrences for each pfam—within eCIS operons, outside eCIS operons, and in total. After counting all pfam occurrences we performed a Fisher Exact Test for pfam enrichment in eCIS operons, compared to the rest of genomes in the analysis. Multiple hypothesis testing correction was performed using the Benjamini-Hochberg procedure. The adjusted $P$ value ($q$ value) of the Core Component was zero, so in order to plot it we changed it to $1e^{-250}$. Then we plotted the data using R enhancedVolcano package[72]. We counted

the number of Phyla each Pfam domain appeared in eCIS. Pfam domains appearing in >10 Phyla were marked as "Core Domains", domains found in 4–10 Phyla were marked as "Shell Domains", and domains found in <4 were marked as "Cloud Domains". These terms were borrowed from pangenomics.

**Toxins and accessory gene clustering; EAT choice criteria**. To identify accessory genes, i.e., genes that are not annotated as homologs of Afp proteins, the operons were expanded from their extrema by four protein coding genes up- and downstream. All genes lacking an annotation of an Afp homolog were clustered. This included the genes up- and downstream of extrema, as well as genes in between annotated Afp homologs. Using CD-HIT version 4.8.1[73], the amino acid sequences of these unmarked genes were clustered with a threshold of at least 40% identity over at least 80% of the length of both query and subject. Each gene was queried by CD-Search[74] (CDD database v3.18) with standard settings to determine if the gene encoded for conserved domains. Each 40%-identity-protein-cluster was then manually examined for domains associated with known biological function, e.g., known toxin domains, known transcription factor domains, etc.

The effector candidates were chosen based on the following criteria: (1) being encoded in the 3′ end of the operon (which is seen commonly in the experimentally studied eCIS), (2) being conserved domain with enzymatic activity, (3) having a domain enriched in eCIS operons (Fig. 3a, Supplementary Data 10), (4) especially genes containing the DUF4157 marker (plus genes next to DUF4157 containing genes).

**Heterologous expression of putative EATs in bacteria**. Candidate gene sequences were retrieved from IMG[16] and were synthesized (codon optimized for *E. coli*), and cloned into either pET28, pET29, or pBAD24 plasmids by Twist Bioscience. The plasmids were designed to express the proteins without a HIS tag, i.e., it was cut from the backbone to prevent interference with potential toxic activity. The plasmids were then transformed by heat shock into *E. coli* BL21 (DE3) pLysS strain. Overnight cultures of the strains harboring the vectors of interest were grown in LB containing the proper selection: Kan or Amp for pET28/29 or pBAD24, respectively. The cultures were normalized to 0.8 OD$_{600}$ and subsequently serially diluted. Dilutions were spotted on LB agar containing the proper selection and inducer (500 μM IPTG, 0.1% arabinose) or repressor (1% glucose) and the plates were incubated overnight at 37 °C.

The toxin-antitoxin assay was done as follows: *E. coli* BL21 (DE3) pLysS strain was transformed with the toxin (EAT 5) on a pET28 plasmid and its predicted antitoxin (IMG gene id 2587734255) on a pBAD24. Two control strains were made, an empty vector (pBAD24) and a truncated antitoxin. We removed the last 51 nucleotides (17 amino acids) from the gene (which was synthesized by TWIST bioscience) using inverse PCR, then it was re-ligated with Nebuilder (NEB) kit. All three strains were tested in a drop assay with three biological replicates. Induction of toxin and antitoxin was performed with 100 μM IPTG and 0.1% arabinose respectively.

**Protein expression patterns of non-toxic candidate EATs**. Fresh *E. coli* BL21 (DE3) pLysS strains with pET28a plasmids were grown overnight in LB (Lennox) supplemented with kanamycin and chloramphenicol. On the following day the overnights were diluted 1:60 in 5 ml cultures and grown to 0.5 OD600 then induced by addition of IPTG (1 mM final concentration) for 3.5 h. OD600 standardize aliquots were taken from each culture, pelleted, resuspended in Laemmli sample buffer and boiled for 15 min. 20 μl of samples where loaded and ran on 12% SDS polyacrylamide gels and stained with Coomassie brilliant blue solution. Gel imaging was carried out using Bio-Rad Chemidoc™ imaging system.

**Heterologous expression of putative toxins in yeast**. Candidate sequences were retrieved from IMG[16]. They were synthesized, following codon adaptation to yeast by Twist Bioscience and were cloned into pESC -leu Galactose inducible plasmids. The plasmids were then transformed into *Saccharomyces cerevisiae* BY4742 strain. Overnight cultures of the strains harboring the vectors of interest were grown in SD -leu media. The cultures OD was normalized to 0.6 nm OD was washed once with water and split into two: one part was grown overnight in SD-leu 2% glucose media and the other part was grown in SD-leu 2% galactose media. Dilutions were spotted on SD-leu plates containing glucose or galactose and the plates were incubated for 48 h at 30 °C.

**Construction of eCIStem repository**. The eCIS database was constructed based on the eCIS data from our research (Supplementary Table 1, Supplementary Data 1–3, 12). The information is divided into four tables: Operons Table, Gene Table, Organism Table and Pfam Table. The eCIStem database scheme is described in Supplementary Fig. 27. The Django framework was used to represent the data and its relationships and deploy it for public usage. eCIStem contains information on "40% Cluster Group" of proteins that share ≥40% identity and ≥80% coverage based on CDHIT[73] clustering. We assume these proteins, that their genes are part of eCIS operons, have similar functionality, so we give them a cluster ID number to mark them as the same.

**S74 Peptidase (PF13884) secondary structure analysis**. We clustered all 84 genes containing PF13884, using CD-HIT[73] version 4.8.1 with 40% similarity. We used two cluster representatives for structural prediction in Phyre2[75], and secondary structure alignment using Praline[76,77].

**Reporting summary**. Further information on research design is available in the Nature Research Reporting Summary linked to this article.

## Data availability
All data are provided through the supplementary tables and eCIStem website. Usage of eCIStem is free for academic usage. Commercial usage of eCIStem requires a permission or a license from Dr. A.L. The Source Data file includes the original images of the EAT drop assays. Source data are provided with this paper.

## Code availability
The code that was used to produce most of the bioinformatics results is described here: https://github.com/alexlevylab/eCIStem.

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

## Acknowledgements

A.L. is generously supported by the Israeli Science Foundation (Grants #1535/20, #3300/20), Alon Fellowship of the Israeli council of higher education, The Hebrew Univeristy - Univeristy of Illinois Urbana-Champain seed grant, and the Israeli Ministry of Agriculture (Grant 12-12-0002). A.M.G. is generously supported by the Kaete Klausner Scholarship and a scholarship from the Israeli Ministry of Aliyah and Integration. We thank Prof. Maya Schuldiner, Prof. Yechiel Shay, and Prof. Saul Burdman for providing different plasmids and cells. We thank Dr. Omri Finkel, Dr. Hila Sberro, Prof. Saul Burdman, Prof. Rotem Sorek, Dr. Erez Mills, Dr. Gal Ofir, and Dr. Lianet Noda-Garcia for careful evaluation of the paper.

## Author contributions

A.M.G. and I.P. contributed equally to this work. A.L. conceived and designed the study. A.L., A.M.G., I.P., K.S. wrote the paper. A.M.G., D.Z. and A.D. performed the computational analysis and designed eCIStem. I.P., K.S. and N.N. performed the molecular biology experiments. W.A., A.L., and A.M.G. designed Deeplasmid. All authors discussed the results and contributed to the final paper.

## Competing interests

The authors declare no competing interests.
