## [Peer Review File · Nature Communications]

REVIEWER COMMENTS

Reviewer #1 (Remarks to the Author):

In the current manuscript, Geller et al use bioinformatics and experimentation to characterize the distribution of extracellular contractile injection systems (eCIS) across prokaryotes. The manuscript dramatically expands the previously known repertoire of eCIS operons, genes, and taxonomic distribution through well-done and thorough well-crafted genomic analyses, including analysis of ecological enrichment. Further, the authors experimentally validate the toxicity of some predicted effector proteins using heterologous expression in bacteria and yeast. The manuscript includes compelling figures and the highly detailed and organized supplemental tables are commendable. Frankly, I think the paper should be accepted, however I think there are a few things that could be added to improve it.

Please use line numbering for ease of reference.

The authors should provide quantitation of toxicity from spot assays via plots, aside from mere images.

Please define "shell" proteins when introduced

The DeepPlasmid methods section should be expanded, as this is a new unpublished tool.

I think Figure 7 (website) is better suited to a supplemental figure.

Use of terms "toxin" and "effectors". Historically, effectors have a strict definition. I appreciate the use of defined toxin domains to identify predicted or putative effectors (and the toxicity data certainly support this), but I feel that it should be made explicitly clear in the text (perhaps in the Discussion) that these proteins are predicted effectors and not experimentally validated to be translocated/injected by the eCIS apparatus.

For EAT toxicity bacterial expression Methods section, confirm that codon optimization was for *E. coli*. Also, confirm if EATs were in-frame with N-terminal His tag, or not.

For the 8 EATs that were not found to be toxic, can you confirm that these proteins are expressed, perhaps by western blotting of lysates with anti-HIS antibodies? If proteins are not expressed well, please indicate this as a caveat to lack of toxicity.

For EAT expression in yeast Methods, "*Cerevisiae*" should not be capitalized. Be sure to italicize genus names (e.g. *Thermus*, in the antitoxin section).

Please make it more explicit in the Figure 8 model which aspects of eCIS function are hypothesized and which have been experimentally demonstrated.

Reviewer #2 (Remarks to the Author):

Geller et al present a bioinformatic analysis of the diversity of eCIS systems across bacteria, with microbiological validation of toxins. eCIS systems are a relatively understudied but potentially important mechanism of bacteria-eukaryote communication and/or attack that does not require direct cell-cell contact. There are many things to be commended in this paper in the wealth of bioinformatic data presented. The online repository of novel eCIS predictions will also be useful to others who wish to survey and verify these systems. However, some of the main conclusions in the paper are unconvincing (see my major comments below). As a result, while the novel predictions may move the field forward somewhat, I do not feel that in its current state the work

represents an advance in understanding likely to influence thinking in the field. This is especially the case given that a bioinformatic survey of eCIS was already published last year in Cell Reports (Chen et al, 2019).

Major

The authors put a lot of weight on their hypothesis that the eCIS is negatively correlated with pathogens of certain animals. What is missing here is an analysis of correlation of eCIS and genome size. Maybe these pathogens have smaller genomes. Maybe in general, the environmental microbes have larger genomes, which would also explain the enrichment? A recent paper in mSystems seems to contradict the observed lack of association of eCIS with bacteria with human hosts (Rojas et al., 2020 DOI: 10.1128/mSystems.00648-20), which should be discussed.

The Blast searches (Blastp presumably - more info is needed on how these were run, with cut-offs etc) should not be used alone to infer relatedness of Afp13 to eukaryotic virus and bacteriophage tail proteins; rather, careful phylogenetic analysis of these hits should be carried out. Hopefully putative phage-like and virus-like Afp13s will be clear in the tree, by their clustering with tail proteins of each type.

I am not convinced that the data shows that eCIS systems have evolved to work against bacteria in some cases. More careful analysis of the tail proteins as suggested above might help the case. While it is clear that several of the identified toxins are toxic in bacteria, that does not rule out that the main target is eukaryotic. The presence of toxin-antitoxins (TAs) at the periphery of eCIS gene clusters is not strong evidence that the toxins are eCIS-delivered effectors targeting bacteria (although the additional presence of the DUF4157 domain may be a good indication?). Rather, the TAs could be addiction modules for whatever mobile region the eCIS is encoded on, i.e. plasmid or, perhaps, transposon. Or for stabilising the chromosomal region in general (TAs can often be found in/near pathogenicity islands, prophages, defence islands etc). Might it be possible to search this dataset of predicted effector sequences to look for a novel signal peptide signature sequence for delivery of an effector by eCIS (whether to bacteria or eukaryotes)? I imagine such a discovery would be very important to the eCIS field, and help in the future prediction of eCISs and their biological functions.

The authors should acknowledge that the 5' and 3' boundaries of the predicted eCIS systems may not be precise. For example, it's very unlikely that the translation factor TypA (642704989) is part of the eCIS, and incorrect boundary prediction may also have led to the inclusion of TAs as potential effectors as described above.

The authors should explain more the relationship of eCIS to Type VI secretion and phages in terms of protein composition, to better understand the rationale for filtering out potential systems with hits to those several pfam domains (what are all those domains?). Could this filtering lead to some false negatives? Is there still risk of false positives? How do the results and strategy compare to (Chen et al, 2019)? Are the same six families that they report found here?

Minor

Page numbers and line numbers would have been very useful.

Methods:

Were the 64,756 genomes limited in some way, e.g. to only bacteria? There should be a list of the genomes considered, preferably annotated with predicted presence/absence of an eCIS system, or the presence of linked core pfam domains. This way people can easily look up their favourite bacteria and see if it is predicted to have an eCIS, directly from the SI.

"only the top half... according to bit score": this needs some clarification.

Make sure to include versions and parameters for all tools, such as HMMSEARCH

inputted isn't a word

It's not clear to me that it makes sense to concatenate Afp8 and Afp11. In fact, this is a bad idea if they may be transferred by HGT separately. If trees are made with the single proteins alone, is the tree topology the same/similar? Does "present together" mean they are encoded adjacent to each other? The sequence alignment should be shared, and it should be stated how many sites were used to make the phylogenetic tree after cutting with TrimAL?

Why generate an UPGMA guide tree for input to scoary? FastTree would be better. In general, IQTree and RaxML are the recommended methods for maximum likelihood phylogenetic analysis. FastTree is great for quick answers, and where the dataset is too large to run with IQTree or RaxML. Bootstrap support values should be calculated and shown.

Please describe the AUC ROC and why it matters.

Custom python scripts should be better explained in terms of what they do. They could also be shared on Github for instance.

Figure 1B: Presumably this is just a subset of genera?

There are no bootstrap support values shown on the trees 1A and 3B, The interesting part is also too small to see. The tree could be enlarged (perfectly ok if a few branches at 7 o'clock extend outside the circle)

"we were surprised by its scarcity in microbial genomes": perhaps the authors could speculate on whether false negatives might be the reason for its apparent scarcity? I.e. if 3 rather than 4 pfam domains were allowed for a potential hit, more could be found. I don't suggest necessarily that the authors repeat their searches with more lenient requirements, just discuss a little whether there might be more distant homologues of eCIS that slip through the net. A related thought is that maybe all photorhabdus genomes get hits because the models were made based on the system in this genus. If the authors remade the Afp8 and Afp11 HMMs with their new hits, they might find more relatives.

Methods: "The toxin-antitoxin assay was done as followed" should be follows.

DUF4157 as a protease domain in a larger toxin-containing protein sounds similar to secreted polymorphic toxins, where the protease domain self-processes the peptide, and releases the toxin domains. Perhaps DUF4157 acts as an immunity domain, which activates the toxin when it self-cleaves to release the toxin?

It seems Fig 3B makes Fig 1A rather redundant. Maybe better to combine, showing plasmids on the same figure?

"some of these domains are likely antibacterial toxins, such as: a DNase..." Presumably this is equally toxic to eukaryotic cells.

EAT7 is annotated as glycoside hydrolase in fig 4B, but "unknown" in table 1. For unknowns, it might be worth trying HHPred. For example, EAT8 is predicted with HHPred to be a nucleoporin.

Footnote of table 1 "S. ceteviciae" typo, and in figure 6 legend "Cerevisiae" should not be capitalised

Figure S5; synteny is misspelled

Reviewer #3 (Remarks to the Author):

Geller et al have carried out a comprehensive, very well designed-and-performed study of the poorly known eCISs. Their bioinformatic work covers an impressive number of 65K genomes including archaea and bacteria where eCISs were found in 1-2% of them among 15 phyla with evidence for HGT. The study opens a range of possibilities for eCIS functions and ecological roles as well as a source of new toxins. It is of interest that eCISs are widely distributed in environmental strains and total absence of any known human pathogen despite being present in other host-associated systems. The authors described for the first time the possibility of eCISs having antibacterial activity and identified new core genomes, accessory genes (regulatory genes) and novel putative effectors and toxin-immunity pairs. The comparative study of tail fibres with viruses from different origins is a curious strategy that provides evidence for putative target cells. The comprehensive in silico work is complemented with experimental data proving the toxicity of a set of newly described effectors in bacteria and/or yeast. Lastly, the authors have created a valuable database available for the scientific community with information about the newly identified eCIS in this work (>1.5K) including gene architecture information, protein, protein domains and metadata information about the microorganisms included.

I have greatly enjoyed reading this work and only have many but minor requests and comments:

- There are 1425 identified eCIS loci in 1249 genomes, these numbers implied that some genomes have more than one eCIS loci but most of them will have only one. The authors comment that *Photobacterium* genomes contain from 2 to 5 eCIS operon per genome. It would be interesting to briefly comment on the distribution of eCIS operons per genome not only in *Photobacterium* but in a more general way, i.e by Phyla/ Genera
- The authors identified a group of eCIS likely targeting bacteria by studying the fibre proteins and also identified clusters with putative antibacterial toxin and/or immunity genes. Have they crossed this information? It would be appropriate to show the clusters that have both attributes indicative of the target cell being bacteria and in a parallel way, the clusters with fibre homologous to "eukaryotic" viruses and the presence of putative eukaryotic effectors in those same clusters.
- Have they found any cluster containing both antibacterial and anti-eukaryotic effectors encoded within the cluster? If so, how is the tail fibre in this case?
- Interestingly, they have found 19 clusters in archaea, anything different at the level of core genes, accessory genes or more importantly toxins? Same with Gram +, most of them are in Gram- bacteria, anything different in the clusters found in Gram+?
- Any hypothesis for the strong enrichment in environmental microbes and depletion from mammalian and avian microbiomes?
- Could data from figure 2 be incorporated to Figure 1? Would it be possible that the ecosystem, lifestyle and host data correlate with the distribution of loci in the phylogenetic tree? Maybe a supplementary figure combining two? Fig 1 and 2 are great and probably to mix them in the main text is not a good idea but I am curious to know the information that this combination could provide
- In the section of the tail fibres, why do they use only 629 Afp13 proteins out of the 1425 eCIS loci? Does it mean that 800 does not have Afp13 or do they were identical to other Afp13 and thus they were not included? They comment the genome with the peptidase S74 don't have Afp13, are there many of this kind?
- Could long domains (1000 amino acids linker) next to DUF4157 be a RHS-similar structure? RHS proteins are described to form a shell-like structure to protect C-terminal toxin domain within the same protein. Any data on these domains predicted 3D structure? That could avoid the need for an antitoxin together with the fact discuss in the discussion that these cells are going to die anyway.
- It is not clear to me how the peptidase S74 would have the same function as the tail fibre
- The authors should clearly state where to find the list of effectors and indicate (colour code for example) in the supplementary file the ones with antibacterial activity and the ones anti-eukaryotic. Also, it should be indicated the ones selected in the study for the experimental section
- How have been the effectors identified? Is it by homology? Do they have any marker domains (N-term signal sequences like T3E or MIX or PAAR domains like T6E? Could it be more? Are there proteins of unknown functions found in the loci 3' that could be potential effectors?
- In the sections where the effectors are described, it should be included the fact that not many genes encoding immunity proteins are found downstream toxin genes. There is a comment in the discussion but it should be included in the data section before the discussion
- In the text "Overall, we identified at least 71 protein domains that are likely toxins and are found

next to eCIS core genes. ", a reference to the table where these proteins are detailed should be included

- For the 20 genes tested as encoding putative toxins, 8 didn't have a toxic effect. There is a possibility that the target was the periplasm and thus the effect is not seen when expressing the putative toxin in the cytosol of E. coli. Including a signal peptide to send the putative effector to the periplasm could fix the problem, at least for some of them
- Are the putatively identified toxins similar to T6SS effectors or other identified polymorphic toxins?
- Table 1 could be sent to supplementary
- EAT13 could be added to Fig 4B
- The example of the toxin-immunity pair of Fig 5 (the RES-like toxin) it is not clear if it was one of the previous described in Fig 4 and if it is not, it would be good to explain why is that. None of Fig 4 has immunity genes? Why was RES-like toxin not tested in Fig 4?
- Some of the putative antibacterial effectors also have anti-eukaryotic effect on yeast. Do these effectors have immunity genes associated? How are the fibres? Could it be that the toxicity in E. coli was unspecific?
- After the text: "We predicted new EATs that target eukaryotes based on their presence in the 3' end of the eCIS operon, presence of eukaryotic domains within the encoded proteins, predicted enzymatic activity on eukaryote-specific molecules (such as actin), and similarity to known virulence factors.", a reference with the table where these effectors are detailed should be added and colour code the selected ones (similar than before for antibacterial effectors).
- "Upon expression induction, we identified four EATs that efficiently killed yeast (Figures 6, S9)." Figure S9 does not correspond here and a comma should be added after "induction"
- The database is a great tool but the "40% Cluster Group" is not clear what does it refers to"
- In the discussion, the authors comment on the fact that there were already 7 EATs identified and that they have added 13 novel ones. Does their methodology allow the identification of the 7-known EATs? As a positive control of how much it covers
- In the discussion, the authors stated: "The fact that most eCIS tail fibers matched both phage and eukaryotic-targeting viruses suggests they may have broad spectrum binding activity, and therefore broad targeting activity." Whereas while reading the results section, they implied that only 18% matched phage or eukaryotic viruses and most didn't match any of them. This point needs to be clarified.
- Figure 1B, genera should be italicised
- The authors should be consistent and use either "fiber" or "fibre", but always the same variant
- In the text: "In some cases, we were able to experimentally identify immunity genes against that rescued bacteria from self-intoxication by their cognate toxins, supporting the toxin intended activity against bacteria". The word "against" is not necessary
- Figure reference in the text does not appear in order, for example, Figure S6 appear before Figure S5 in the main text, it is not the only one (Fig S8 before S7), double-check all of them
- Table 1 – errata DUF4147 instead of DUF4157
- Fig 5. B. XRE-like Antitoxin Protects From RES-like Toxin – no need to capitalise every first letter of every word
- In the text: "In one case, The toxin, EAT5, resembles RES (Figure 4b), an NADase that is accompanied by an antitoxin called Xre50,51." T from The should not be capitalised.
- The text: "Following our extensive bioinformatic analysis we were also interested in discovery of novel eCIS anti-eukaryotic toxins." Would read better like this: "we were also interested in discovering novel eCIS ..."
- After the text: "EAT13 gene was tested only in yeast. These are predicted to act as an ADP-ribosyl transferase, a chitinase, a deaminase, and an actin cross-linker", the word "respectively" should be added
- In the text: " We identified that some eCIS tail fibres that can help predict whether eCIS target bacteria or eukaryotes.", the second "that" after "fibres" should be eliminated

All changes in the paper as a result of the review process appear in blue.

REVIEWER COMMENTS

Reviewer #1 (Remarks to the Author):

In the current manuscript, Geller et al use bioinformatics and experimentation to characterize the distribution of extracellular contractile injection systems (eCIS) across prokaryotes. The manuscript dramatically expands the previously known repertoire of eCIS operons, genes, and taxonomic distribution through well-done and thorough well-crafted genomic analyses, including analysis of ecological enrichment. Further, the authors experimentally validate the toxicity of some predicted effector proteins using heterologous expression in bacteria and yeast. The manuscript includes compelling figures and the highly detailed and organized supplemental tables are commendable. Frankly, I think the paper should be accepted, however I think there are a few things that could be added to improve it.

Thank you for close reading of the manuscript and for your support for publication. We address below your comments and suggestions.

Please use line numbering for ease of reference.

We added line numbers as requested.

The authors should provide quantitation of toxicity from spot assays via plots, aside from mere images.

We have now quantified the toxicity, and added a graph of the results (Figure S21). In one case, EAT6, there was little difference in terms of CFU, but the colonies received upon EAT6 induction are very small (Figure S20).

Please define “shell” proteins when introduced

We defined it on line 214 (“we defined ‘shell’ domains as those that are present in eCIS from 4-10 microbial phyla and likely produce specific eCIS subtypes or are responsible for regulation”) and in the legend of Figure 3.

The DeepPlasmid methods section should be expanded, as this is a new unpublished tool.

We expanded the description of Deepplasmid method and its application under the M&M section titled “Detection of eCIS within plasmids using Deepplasmid”.

I think Figure 7 (website) is better suited to a supplemental figure.

It is now moved to the supplemental section (Figure S26).

Use of terms “toxin” and “effectors”. Historically, effectors have a strict definition. I appreciate the use of defined toxin domains to identify predicted or putative effectors (and the toxicity data certainly support this), but I feel that it should be made explicitly clear in the text (perhaps in the Discussion) that these proteins are predicted effectors and not experimentally validated to be translocated/injected by the eCIS apparatus.

We have updated our use of “effector” as per this comment. In the introduction, we define the term “effector”, and use it to describe experimentally proven effectors. In the results section, we use the eCIS-associated toxin (EAT) acronym, and do not call them effectors.

For EAT toxicity bacterial expression Methods section, confirm that codon optimization was used for E. coli. Also, confirm if EATs were in-frame with N-terminal His tag, or not.

We changed the methods to confirm that indeed the sequences of the putative toxins were codon optimized for *E. coli*, and that they were designed to remove the HIS tag. We did the latter in order to lower the chances that a tag, albeit a small one, would affect any activity of the putative toxin.

For the 8 EATs that were not found to be toxic, can you confirm that these proteins are expressed, perhaps by western blotting of lysates with anti-HIS antibodies? If proteins are not expressed well, please indicate this as a caveat to lack of toxicity.

We tested protein expression of failed EATs.

We added the following text:

Line 267: One candidate EAT gene failed cloning, likely due to high toxicity even without expression induction (Table S13).

Line 272: We tested for protein expression of five of the predicted EATs that were not found to be toxic. We saw that three of these were expressed and are therefore indeed non-toxic to *E. coli* and two were not detected and therefore their heterologous expression may have been impaired (Figure S22, Table S13).

*For EAT expression in yeast Methods, “Cerevisiae” should not be capitalized. Be sure to italicize genus names (e.g. *Thermus*, in the antitoxin section).*

These issues were fixed.

Please make it more explicit in the Figure 8 model which aspects of eCIS function are hypothesized and which have been experimentally demonstrated.

Figure 7 (previously Figure 8) now contains “?” to denote aspects of eCIS function that are predicted based on our results yet have not been experimentally confirmed.

Thanks again for a very thorough review!

Reviewer #2 (Remarks to the Author):

Geller et al present a bioinformatic analysis of the diversity of eCIS systems across bacteria, with microbiological validation of toxins. eCIS systems are a relatively understudied but potentially important mechanism of bacteria-eukaryote communication and/or attack that does not require direct cell-cell contact. There are many things to be commended in this paper in the wealth of bioinformatic data presented. The online repository of novel eCIS predictions will also be useful to others who wish to survey and verify these systems. However, some of the main conclusions in the paper are unconvincing (see my major comments below). As a result, while the novel predictions may move the field forward somewhat, I do not feel that in its current state the work represents an advance in understanding likely to influence thinking in the field. This is especially the case given that a bioinformatic survey of eCIS was already published last year in Cell Reports (Chen et al, 2019).

We thank Reviewer #2 for reading the manuscript carefully and providing their critical comments that led to improvement of the resulting manuscript, in our view. We cite Chen et al. pioneering work in the field and use their results in our present study, which expands upon their work. Their work indeed also bioinformatically identified eCIS operons and built a database (dbeCIS). However, beyond this, there is little similarity. Our bioinformatic analysis expanded further to look at statistical enrichments and depletions of (1) phylogenetic taxa, (2) environment and lifestyle, and (3) pfam domains associated with eCIS. Most importantly, a bulk of our study is also concerned with prediction of, and many experiments with predicted toxic effector proteins, which are not investigated at all by Chen et al., neither bioinformatically nor experimentally. We provide genomic and experimental data suggesting that eCIS may target prokaryotic cells, which is unprecedented. We respond to your major and minor comments below and hope that our responses help address your concerns.

Major

The authors put a lot of weight on their hypothesis that the eCIS is negatively correlated with pathogens of certain animals. What is missing here is an analysis of correlation of eCIS and genome size. Maybe these pathogens have smaller genomes. Maybe in general, the environmental microbes have larger genomes, which would also explain the enrichment? A recent paper in mSystems seems to contradict the observed lack of association of eCIS with bacteria with human hosts (Rojas et al., 2020 DOI: 10.1128/mSystems.00648-20), which should be discussed.

We point out that in the Rojas et al. paper, they find that eCIS are encoded in *Bacteroidales*, and these eCIS (called "BIS") are found in human microbiomes. We clarify that we do not claim that eCIS are absent from human microbiomes, rather we say they are statistically depleted in human microbiomes. To better visualize this, we did an analysis to complement Figure 2, that shows metadata distribution, rather than

enrichment/depletion, where we see bacterial strains encoding for eCIS that have been isolated from humans (Figure S8, Panel B). Secondly, we show not only that eCIS are present in *Bacteroidales* order, but show eCIS are enriched in genera of this order. For example, we see a modest enrichment of the genus *Bacteroides*, and a strong enrichment of the genus *Parabacteroides*. Therefore, our data does not contradict Rojas et al., rather we put their very interesting results into a broader context: although eCIS is indeed found in humans, it may be an exception rather than the rule.

You raise an interesting point regarding the negative correlation between eCIS and pathogens. We performed an analysis to check for a correlation between eCIS and overall genome size. Although we cannot logically give primacy to either correlation, we do see that indeed genomes encoding eCIS tend to be larger (Figure S9) Combined with the results shown in Figure 2, we interpret this to mean that bigger genomes are indeed normally associated with eCIS, as you suggest. We note, although, that small genome endosymbionts are found to contain eCIS as well, for example, the *Candidatus Regiella insecticola LSR1*, which harbours an eCIS even though its genome size is ~ 2Mbps and contains less than 2000 genes. This analysis is described in lines 151-154.

The Blast searches (Blastp presumably - more info is needed on how these were run, with cut-offs etc) should not be used alone to infer relatedness of Afp13 to eukaryotic virus and bacteriophage tail proteins; rather, careful phylogenetic analysis of these hits should be carried out. Hopefully putative phage-like and virus-like Afp13s will be clear in the tree, by their clustering with tail proteins of each type.

To clarify the methods we used, we changed the wording in the Materials and Methods section titled "Systematic tail fiber (Afp13) sequence analysis", and also produced Figures S11-14. Figure S12 shows specific Afp13 genes (X-axis) and bitscore of their best hit against Eukaryotic virus database subtracted from their best hit from the tailed bacteriophage database. Figure S13 shows the same thing for S74_peptidase genes, which we believe also may play a role as an eCIS tail fiber, as described in the text.

We then made a maximum likelihood tree (Figure S11) that shows the phylogenetic relationship between Afp13 genes. Although there is no clear pattern, we can see areas where phage-like tails dominate. Interestingly, we see in Figure S14 that S74_peptidase containing genes are those that are overwhelmingly phage-like. Notably, most Afp13 proteins had no significant hits vs. either database, meaning they may have a unique structure, which needs to be further investigated.

To get another point of view on this, we also plotted all the Afp13 blast results onto the phylogenetic tree of eCIS-encoding strains (constructed based on Afp8 and Afp11), and we see an interesting clustering of Eukaryotic virus-like Afp13 in the area of the tree with the known eCIS (Afp, PVC, MACS), that are known to infect eukaryotes (Figure S14).

I am not convinced that the data shows that eCIS systems have evolved to work against bacteria in some cases. More careful analysis of the tail proteins as suggested above might help the case. While it is clear that several of the identified toxins are toxic in bacteria, that does not rule out that the main target is eukaryotic. The presence of toxin-antitoxins (TAs) at the periphery of eCIS gene clusters is not strong evidence that the toxins are eCIS-delivered effectors targeting bacteria (although the additional presence of the DUF4157 domain may be a good indication?). Rather, the TAs could be addiction modules for whatever mobile region the eCIS is encoded on, i.e. plasmid or, perhaps, transposon. Or for stabilising the chromosomal region in general (TAs can often be found in/near pathogenicity islands, prophages, defence islands etc). Might it be possible to search this dataset of predicted effector sequences to look for a novel signal peptide signature sequence for delivery of an effector by eCIS (whether to bacteria or eukaryotes)? I imagine such a discovery would be very important to the eCIS field, and help in the future prediction of eCISs and their biological functions.

First, we agree that an experimental validation is required to demonstrate direct targeting of bacteria beyond EAT toxicity in heterologous expression. We modified our final model (Figure 7) and added a question mark for all aspects that await these experimental results, with a focus on the bacterial targeting. We agree that some toxin-antitoxin systems might be part of the mobile element containing the eCIS.

However, as the reviewer mentioned, in several cases the toxin of the TA pair is fused to a DUF4157 domain which is an eCIS core domain, suggesting strongly these toxins are connected to the eCIS operon. In addition, these data are added to the tail fibre analysis that demonstrate high similarity of Afp13 to phage tails in some instances. Moreover, 5 EATs were toxic to E. coli but not to yeast (data added to Table 1)

Finally, we have added the results of a new experiment (Figure S23), showing that EAT10 toxicity is enhanced with addition of a twin arginine sequence (TAT) that translocates it to the periplasm. In fact, EAT10 in a pBAD backbone is not toxic (as opposed to in a pET28 vector; Figure S23 vs. Figure 4), and addition of the TAT makes it toxic. EAT10 in the periplasm is very toxic, which is consistent with the fact that it is predicted to be a peptidoglycan hydrolase (Figure 4B).

We are currently looking for a signature signal peptide for delivery by eCIS.

The authors should acknowledge that the 5' and 3' boundaries of the predicted eCIS systems may not be precise. For example, it's very unlikely that the translation factor TypA (642704989) is part of the eCIS, and incorrect boundary prediction may also have led to the inclusion of TAs as potential effectors as described above.

This is a good point, and we added some of the following information to the materials and methods, in order to make this clear. The eCIS has inherently varied boundaries because (1) we see that many times the 3' end has toxins, which, according to our “cloud” gene analysis, are quite varied in terms of their domains, and therefore this causes a variation in the 3' region; (2) we see evidence of horizontal gene transfer, thereby creating varied genes surrounding the eCIS operon, i.e. whatever genes happened to be surrounding the insertion site of the eCIS; (3) core genes sometimes have multiple copies, which increases the size of the eCIS operon. All these considerations makes it hard to define boundaries *in silico*. We decided to pick, arbitrarily, four genes down and upstream of the first and last Afp homologs in the operon. It is a rough estimation, and likely has both false positives and false negatives. The best case scenario would be to use transcriptional information to understand exactly which genes need to be expressed in order to build a functional eCIS or genetics combined with microscopy to detect the essential genes for eCIS production. Unfortunately, we do not have this information, and we acknowledge that we have “fuzzy” boundaries for our operons.

The authors should explain more the relationship of eCIS to Type VI secretion and phages in terms of protein composition, to better understand the rationale for filtering out potential systems with hits to those several pfam domains (what are all those domains?). Could this filtering lead to some false negatives? Is there still risk of false positives? How do the results and strategy compare to (Chen et al, 2019)? Are the same six families that they report found here?

We have organized the pfams and COGs used in each step of filtering in a new supplementary table (Table S2) along with the summary of the domain, in order to make it clear why each pfam was used, and what each domain is, beyond its accession code. We added some examples to the Materials and Methods, as well. The phage domains chosen are domains involved in capsid structure, or in replication/transcription, which eCIS do not have, as they are “headless” and do not carry genetic material, as far as science is aware. The T6SS domains were specifically chosen along the same lines, e.g. COG3523, which is found in TssM, a membrane complex protein absent in eCIS.

We added information regarding the false positive / false negative ratio to the Materials and Methods. In short, our algorithm tended to be stringent, i.e. tended to minimize false positives, at the risk of possible false negatives. We reason that the high evolutionary similarity between eCIS and other related systems could lead to erroneous conclusions in downstream analysis, so we opted for fewer eCIS, but with more certainty that they are indeed bona fide eCIS.

Chen et al. grouped eCIS into subclades (Ia, Ib, IIa, IIb, IIc, IId). To compare our database with theirs, we took eCISem *Afp8* and *Afp11* genes, and many *Afp8* and *Afp11* genes from the Chen et al. study, and created a maximum likelihood tree of a combination of the datasets. Figure S5 shows that the genes from our study and Chen et al.'s work nicely interweave on the tree. We identify eCIS related to the subclades they found, but we find a different relationship between them. Namely: the division to claded I and II was detected by our analysis too but not the division within clade II. We

also made Figure S3 to show the bootstrap values above 0.8 on this tree, which may explain the differences in tree topology. We conclude that although we used a different database (IMG) than Chen et al. (NCBI), we were able to find results that are consistent with those that Chen et al. found.

Minor

Page numbers and line numbers would have been very useful.

We added them to the newest draft.

Methods:

Were the 64,756 genomes limited in some way, e.g. to only bacteria? There should be a list of the genomes considered, preferably annotated with predicted presence/absence of an eCIS system, or the presence of linked core pfam domains. This way people can easily look up their favourite bacteria and see if it is predicted to have an eCIS, directly from the SI.

We added a table, as recommended (Table S1)

"only the top half... according to bit score": this needs some clarification.

This was reworded to be more clear: "Hmsearch results were ranked by bitscore, and we only considered results whose bitscore was in the top half of all scores, i.e. we dropped the bottom 50% of hits, in order to minimize false positive hits."

Make sure to include versions and parameters for all tools, such as HMMSEARCH

We changed the Materials and Methods to be more specific with versions and parameters.

inputted isn't a word

Noted. This word was removed, and the phrasing was changed.

It's not clear to me that it makes sense to concatenate Afp8 and Afp11. In fact, this is a bad idea if they may be transferred by HGT separately. If trees are made with the single proteins alone, is the tree topology the same/similar? Does "present together" mean they are encoded adjacent to each other? The sequence alignment should be shared, and it should be stated how many sites were used to make the phylogenetic tree after cutting with TrimAL?

By concatenating Afp8 and Afp11, we aimed to add more information than just one core

gene in building our phylogeny. The same reasoning showed that concatenation of multiple single copy marker genes yields a more accurate tree than using only a single gene tree.

We do not believe the two genes are transferred separately by HGT. Rather, we imagined a model where the genes in the operon together make up a “functional unit”. As known from the literature, AFP is encoded on a plasmid. We also predict that eCIS are encoded on plasmids (Figure 1, S2). This fact supports the idea that eCIS operons are functional units, where core elements are horizontally transferred together on the same piece of plasmid DNA. Since we only looked at linked Afp8 and Afp11 genes in operons (rather than orphan genes) we claim that the most parsimonious explanation suggests that they are transferred together.

To confirm this idea, we have taken your suggestion to plot the *afp8* and *afp11* alone, and we indeed see a largely similar topology between them (Figure S4).

“Present together” means in the same operon, not necessarily one next to the other. The wording was changed to clarify this point.

Afp8 had 1632 occurrences in our database, and after trimming, all 1632 remained. Afp11 had 1601 occurrences in our database, and after trimming, 1587 remained. In a single operon, *afp8* and *afp11* co-occurred 1407 times; thus, 1407 sequences built the tree.

Dataset S1 contains the sequence alignment of Afp8+Afp11 sequence alignment.

Why generate an UPGMA guide tree for input to scoary? FastTree would be better. In general, IQTree and RaxML are the recommended methods for maximum likelihood phylogenetic analysis. FastTree is great for quick answers, and where the dataset is too large to run with IQTree or RaxML. Bootstrap support values should be calculated and shown.

The original Scoary method (Brynildsrud O et al., *Genome Biol.* 2016;17:238) takes in pan-genomes of bacteria, and, by using a matrix of gene presence and absence, calculates, based on Hamming distance, a UPGMA tree. This tree is used to correct for phylogenetic artefacts in enrichment analyses. We have modified the inputs in order to check if eCIS presence explains enrichment of metadata categories. Because this “microbial GWAS” method relies on the bacteria being closely related, and uses gene presence and absence as a proxy for phylogenetic proximity, we see your question as a very good point, as indeed the 16s tree may not be ideally constructed by the relatively simple UPGMA algorithm.

Therefore, we have rerun the Scoary analysis using a maximum likelihood tree 16s rRNA tree constructed with FastTree, and we see similar results (Figure 2, Table S8).

Please describe the AUC ROC and why it matters.

In the M&M section we expanded on unpublished Deeplasmid algorithm and explained about AUC ROC (line 490): "Deeplasmid achieves an AUC-ROC of over 93% on a separate dataset of ~6,000 sequences from IMG. The AUC - ROC is a performance measurement for classification at various thresholds settings. AUC represents how well the model is capable of distinguishing between classes. The ROC curve is plotted as the true-positive rate (TPR) against the false-positive rate (FPR) where TPR is on y-axis and FPR is on the x-axis. A higher AUC means the model is better at predicting chromosomes as chromosomes and plasmids as plasmids."

Custom python scripts should be better explained in terms of what they do. They could also be shared on Github for instance.

We added a Github <https://github.com/alexlevylab/eCISem> and a link to it at the top of the Materials and Methods section (line 415).

Figure 1B: Presumably this is just a subset of genera?

Yes, we chose some for display based on how widely known they are. We updated the figure legend to make this clear.

There are no bootstrap support values shown on the trees 1A and 3B, The interesting part is also too small to see. The tree could be enlarged (perfectly ok if a few branches at 7 o'clock extend outside the circle)

We added Figure S3 to show bootstrap values.

We also added supplementary figures with enlargement as described, for easier viewing of the rings versus tree branches (Figure S2).

"we were surprised by its scarcity in microbial genomes": perhaps the authors could speculate on whether false negatives might be the reason for its apparent scarcity? I.e. if 3 rather than 4 pfam domains were allowed for a potential hit, more could be found. I don't suggest necessarily that the authors repeat their searches with more lenient requirements, just discuss a little whether there might be more distant homologues of eCIS that slip through the net. A related thought is that maybe all photorhabdus genomes get hits because the models were made based on the system in this genus. If the authors remade the Afp8 and Afp11 HMMs with their new hits, they might find more relatives.

The eCIS that have been experimentally studied are all encoded in Proteobacteria, specifically in Gammaproteobacteria. This comment suggests that because the models were built on these few eCIS, we perhaps may expect to get a narrow band of similar microbes. However, we see a wide distribution of taxonomic groups, even to the point of

two kingdoms, Bacteria and Archaea. We take this to believe that our results do not reflect a bias.

Notwithstanding, there is a possibility that there are certain eCIS clades that are different than the one mentioned in the paper and share only a subset of the eCIS core genes. In a similar fashion different subtypes of T6SS were identified in recent years. Our approach was very stringent as mentioned in the M&M section to avoid erroneous classification of other things (pyocins, T6SS) as eCIS which may reduce the reliability of this work.

Methods: "The toxin-antitoxin assay was done as followed" should be follows.

Changed.

DUF4157 as a protease domain in a larger toxin-containing protein sounds similar to secreted polymorphic toxins, where the protease domain self-processes the peptide, and releases the toxin domains. Perhaps DUF4157 acts as an immunity domain, which activates the toxin when it self-cleaves to release the toxin?

This is a great idea! We could not detect a conserved cleavage site. We work on elucidating DUF4157 function from both molecular and computational approaches.

It seems Fig 3B makes Fig 1A rather redundant. Maybe better to combine, showing plasmids on the same figure?

We considered this option, but we did not want it to be too intimidating, so we believe it may be better as two separate figures, especially given how many data layers Figure 3B already has. We believe it may be easier to digest as two separate figures.

"some of these domains are likely antibacterial toxins, such as: a DNase..." Presumably this is equally toxic to eukaryotic cells.

We agree and we deleted this sentence.

EAT7 is annotated as glycoside hydrolase in fig 4B, but "unknown" in table 1. For unknowns, it might be worth trying HHPred. For example, EAT8 is predicted with HHPred to be a nucleoporin.

We fixed the inconsistency between Figure 4B and Table 1.

Footnote of table 1 "S. ceteviciae" typo, and in figure 6 legend "Cerevisiae" should not be capitalised

Both are fixed.

Figure S5; synteny is misspelled

Fixed.

Thanks again for a very thorough review!

Reviewer #3 (Remarks to the Author):

Geller et al have carried out a comprehensive, very well designed-and-performed study of the poorly known eCISs. Their bioinformatic work covers an impressive number of 65K genomes including archaea and bacteria where eCISs were found in 1-2% of them among 15 phyla with evidence for HGT. The study opens a range of possibilities for eCIS functions and ecological roles as well as a source of new toxins. It is of interest that eCISs are widely distributed in environmental strains and total absence of any known human pathogen despite being present in other host-associated systems. The authors described for the first time the possibility of eCISs having antibacterial activity and identified new core genomes, accessory genes (regulatory genes) and novel putative effectors and toxin-immunity pairs. The comparative study of tail fibres with viruses from different origins is a curious strategy that provides evidence for putative target cells. The comprehensive in silico work is complemented with experimental data proving the toxicity of a set of newly described effectors in bacteria and/or yeast. Lastly, the authors have created a valuable database available for the scientific community with information about the newly identified eCIS in this work (>1.5K) including gene architecture information, protein, protein domains and metadata information about the microorganisms included.

I have greatly enjoyed reading this work and only have many but minor requests and comments:

We are glad you enjoyed reading the manuscript, and we have addressed your concerns below.

- *There are 1425 identified eCIS loci in 1249 genomes, these numbers implied that some genomes have more than one eCIS loci but most of them will have only one. The authors comment that Photorhabdus genomes contain from 2 to 5 eCIS operon per genome. It would be interesting to briefly comment on the distribution of eCIS operons per genome not only in Photorhabdus but in a more general way, i.e by Phyla/ Genera*

We have added an additional supplementary table (Table S7) listing all genomes containing multiple eCIS operons, along with their count. We added the text: "Interestingly, 146 genomes, mostly from *Photorhabdus*, *Dickeya*, *Actinokineospora*, *Streptomyces*, *Algoriphagus*, *Chitinophaga*, *Flavobacterium*, and *Calothrix* genera, were

found to contain more than one eCIS operon, ranging from two to five copies per genomes (Table S7)."

- *The authors identified a group of eCIS likely targeting bacteria by studying the fibre proteins and also identified clusters with putative antibacterial toxin and/or immunity genes. Have they crossed this information? It would be appropriate to show the clusters that have both attributes indicative of the target cell being bacteria and in a parallel way, the clusters with fibre homologous to "eukaryotic" viruses and the presence of putative eukaryotic effectors in those same clusters.*

We added information to Table 1 denoting if an Afp13 gene is in the operon. We do see one case where the phage-like tail is indeed associated with an antibacterial EAT (EAT 5), and three cases where the Afp13 does not have a strong homology to a known tail fibre (EAT 1, 3, and 10). Because of the paucity of operons that actually encode for Afp13, these are the only four of our EATs which are associated with Afp13. We therefore cannot draw any large conclusions.

- *Have they found any cluster containing both antibacterial and anti-eukaryotic effectors encoded within the cluster? If so, how is the tail fibre in this case?*

Although we did not do a systematic analysis to computationally identify the target cells of effectors, we do see that EAT1, EAT3, and EAT11 can kill both prokaryotes and eukaryotes (Figure 4 and 6). In all three cases we could not find a clear hit towards virus or phage. This information was added to Table 1.

- *Interestingly, they have found 19 clusters in archaea, anything different at the level of core genes, accessory genes or more importantly toxins? Same with Gram +, most of them are in Gram- bacteria, anything different in the clusters found in Gram+?*

We searched for a signature pfam domain per group (Archaea, Gram positive, Gram negative), and we found that there are many pfams that are unique for each group (Figure S18). 26 pfam domains were found that are shared between all three groups, and, as we expected, these pfam domains contain the eCIS core pfam domains.

Although many pfams were unique to each group, we noticed upon closer inspection that these "signature" pfam domains were not a function of group membership (Archaea, Gram positive or Gram negative). For example, in Bacteria (both Gram positive and negative), the most abundant unique pfam for each group was found in less than 10% of the group, meaning that less than 10% of the genomes in the group contained the most highly abundant unique pfam. This suggests that there is no unique pfam signature for each group, and is rather a consequence of variance in a more specific taxonomic level, e.g. the genus, or strain level.

The updated text reads: "Additionally, we looked for a unique pfam domain distribution between Archaea, Gram-positive and Gram-negative bacteria (Figure S18). Although many unique pfam domains can be found in each of the groups, none of them is highly

specific and characteristic of a single taxonomic group. We hypothesize that the unique pfams are a consequence of variability in lower taxonomic levels."

• *Any hypothesis for the strong enrichment in environmental microbes and depletion from mammalian and avian microbiomes?*

We have a few thoughts and possible directions regarding why there is such a strong enrichment in environmental microbes, and a depletion from mammalian and avian microbiomes. First, Chen et al. noted that in their eCIS study a little overlap between microbes encoding eCIS with those encoding T6SS, and they speculate this could mean they have redundant function. It is possible that our results point to the hypothesis that indeed the T6SS is the "preferred weapon" of bacteria that are found in mammalian and avian tissues (including in pathogens), while eCIS is the "preferred weapon" of microbes in the environment. Expanding on this thought, it is interesting to note that T6SS are contact-dependent, while eCIS are extracellular. Diffusion of an eCIS particle may be more of a viable strategy in the water (we found enrichment of eCIS in aquatic environments, see Fig3 and S8A). Indeed, the bacterium that encodes for MACs dwells in the ocean. Although we do not know exactly what triggers the expression of MACs. It seems to be expressed in a small fraction of bacteria in a variety of seawater-mimicing media (Shikuma et al., 2014), supporting the idea that eCIS are simply released into the ocean water for diffusion. Of course, this is speculative, and only part of the story, and needs to be tested further with both bioinformatic and experimental tools.

Another explanation that we propose: If eCIS targets mostly specific eukaryotes (e.g. invertebrates) and to a lesser extent bacteria, we would not expect it to be found in environments where these eukaryotes are absent, such as the human gut and skin.

• *Could data from figure 2 be incorporated to Figure 1? Would it be possible that the ecosystem, lifestyle and host data correlate with the distribution of loci in the phylogenetic tree? Maybe a supplementary figure combining two? Fig 1 and 2 are great and probably to mix them in the main text is not a good idea but I am curious to know the information that this combination could provide*

Yes, we added Figure S8, which displays some of the most important metadata for this study on the eCIS phylogenetic tree. We can see that ecosystem-related metadata such as "environmental", "terrestrial", "aquatic", are widespread across the tree. We also see that the area of the tree with microbes isolated from invertebrates, protists, and nematodes is the very same region that has leaves of the previously studied eCIS, that are indeed associated with invertebrates and nematodes. We can also see in Panel B the paucity of microbes labeled as pathogens, mammals, and human isolated.

• *In the section of the tail fibres, why do they use only 629 Afp13 proteins out of the 1425 eCIS loci? Does it mean that 800 does not have Afp13 or do they were identical to other Afp13 and thus they were not included?*

Afp13 is sparse in our database, i.e. many eCIS operons do not encode AFP13; There are only 371 AFP13 genes in our original dataset. We used a relaxed parameter, which

is described at the Materials and Methods section, to actually increase the amount of AFP13 to 629 genes, to increase the chance we find interesting information that otherwise may have been lost due to false negatives. We added a comment in the results section that in most operons we could not detect an Afp13 gene homolog. We also analyzed S74 genes (n = 98), and see that they look mostly like phage tails.

- *Could long domains (1000 amino acids linker) next to DUF4157 be a RHS-similar structure? RHS proteins are described to form a shell-like structure to protect C-terminal toxin domain within the same protein. Any data on these domains predicted 3D structure? That could avoid the need for an antitoxin together with the fact discuss in the discussion that these cells are going to die anyway.*

This is a very intriguing idea. Our naive thought is that effectors are loaded into the tube of eCIS, as Ericson et al. (eLife 2019;8:e46845) demonstrated for MACs. However, this does not discount the hypothesis that DUF4157 may be some kind of physical shell, and possibly loaded differently onto the eCIS particle (perhaps by interaction with the spike, by analogy to VgrG in the Type VI Secretion System). However, we have not successfully modeled the DUF4157 structure (we tried some modeling with the state-of-the-art methods that are publicly available; Phyre2, amongst others), and it has yet to be solved experimentally. We hope to either come to the answer by solving the structure, or predicting the structure with newer machine-learning based tools that are emerging. The discussion section currently reads: "Another option is related to the yet mysterious function of the DUF4157 domain which accompanies many of the EATs. This domain may confer some immunity to the eCIS producers through temporal or spatial toxin inactivation."

- *It is not clear to me how the peptidase S74 would have the same function as the tail fibre*

This domain is a chaperone domain covalently attached to T5 tail fibers. It helps the assembly and exposure of the C-terminal tail fiber domain in T5 fibers, which is critical for proper function and mature structure. Concerning S74, we don't really know exactly how it functions, but we found that operons that contain this gene lacked an annotated Afp13 (Figure S14), and that this gene was mostly found near Afp12 or Afp14 genes in the operon. We therefore, we hypothesize it might have a similar function in eCIS fibers.

- *The authors should clearly state where to find the list of effectors and indicate (colour code for example) in the supplementary file the ones with antibacterial activity and the ones anti-eukaryotic. Also, it should be indicated the ones selected in the study for the experimental section*

We added Table S13 that includes all the genes we tested for toxicity. Table 1 describes those that demonstrated toxicity against bacteria or yeast.

- *How have been the effectors identified? Is it by homology? Do they have any marker*

domains (N-term signal sequences like T3E or MIX or PAAR domains like T6E? Could it be more? Are there proteins of unknown functions found in the loci 3' that could be potential effectors?

The effector candidates were chosen based on the following criteria: (1) being encoded in the 3' end of the operon (which is seen commonly in the experimentally studied eCIS), (2) being conserved domain with enzymatic activity, (3) having a domain enriched in eCIS operons (Figure 3A), (4) especially genes containing the DUF4157 marker (plus genes next to DUF4157 containing genes). All this information has been added to the Materials and Methods section to clarify (lines 541-544). Except for DUF4157, we have yet to identify a putative marker domain/motif, although a search for one is ongoing. Based on our high success rate in experimental validation we deduce that there is a wealth of possible effectors in these operons, and we will continue to investigate what they have in common, and how we can refine our ability to identify them.

- In the sections where the effectors are described, it should be included the fact that not many genes encoding immunity proteins are found downstream toxin genes. There is a comment in the discussion but it should be included in the data section before the discussion*

Text was changed to include, "Although many of the tested EATs above puzzlingly do not have putative immunity genes associated with them", on line 295.

- In the text "Overall, we identified at least 71 protein domains that are likely toxins and are found next to eCIS core genes. ", a reference to the table where these proteins are detailed should be included*

This information is found in Table S11. The text now reads (Lines 253-255): "Overall, we identified at least 71 protein domains that are likely toxins and are found next to eCIS core genes and compiled a list of all genes that contain at least one of these putatively toxic domains (Table S10-S11). "

- For the 20 genes tested as encoding putative toxins, 8 didn't have a toxic effect. There is a possibility that the target was the periplasm and thus the effect is not seen when expressing the putative toxin in the cytosol of E. coli. Including a signal peptide to send the putative effector to the periplasm could fix the problem, at least for some of them*

Indeed for EAT10 we identified that addition of signal peptide increased toxicity. The results now contain this sentence (lines 275-280): "EAT10 was predicted to function as a peptidoglycan hydrolase. Therefore, we hypothesized that it should be more toxic to E. coli upon expression in the periplasm than in the cytoplasm. We cloned EAT10 with or without the twin-arginine leader motif that translocates proteins into the periplasm and indeed observed a 104 higher toxicity in the periplasm than in the cytoplasm

(Figure S23). This result provides yet another support for potential EAT antibacterial activity since the periplasm is a bacteria-specific compartment."

For other EATs that we tested, including those that failed, we did not see this effect.

- *Are the putatively identified toxins similar to T6SS effectors or other identified polymorphic toxins?*

A BLASTP search of SecReT6 database of T6SS effectors (J. Li et al., Environmental Microbiology, 2015 Jan 30. doi: 10.1111/1462-2920.12794.) for homologs of the experimentally tested EATs did not result in any hits.

We do see, though, that some of the putative antibacterial toxins with an N-terminal DUF4157 domain do have known antibacterial pfam domains in their C-termini. Some of these C-terminal domains indeed are found in T6SS effectors, such as Ntox15 (pfam1560), and Endonuclea_NS_2 (pfam13930). This is expected, given the "interchangeable parts" model of polymorphic toxin systems, and supports the idea the DUF4157 may be a trafficking domain for the eCIS system.

- *Table 1 could be sent to supplementary*

Although it could perhaps be sent to the supplementary, we believe Table 1 is critical as it describes all the information on the EATs, one of the main findings of our study.

- *EAT13 could be added to Fig 4B*

We decided to add an EAT13 cartoon describing its encoding operon and protein architecture, as suggested, but with Figure 6, so it is together with the cognate drop assay.

- *The example of the toxin-immunity pair of Fig 5 (the RES-like toxin) it is not clear if it was one of the previous described in Fig 4 and if it is not, it would be good to explain why is that. None of Fig 4 has immunity genes? Why was RES-like toxin not tested in Fig 4?*

The RES-like toxin is EAT5, which is indeed described in Figure 4. We added in the legend of Figure 5 a note that we refer to EAT5. The legend of Figure 4 also denotes the EAT5 adjacent gene as predicted immunity protein.

- *Some of the putative antibacterial effectors also have anti-eukaryotic effect on yeast. Do these effectors have immunity genes associated? How are the fibres? Could it be that the toxicity in E. coli was unspecific?*

We didn't check experimentally, but bioinformatically, we did not see genes annotated as immunity proteins associated with these genes. This is one of the reasons why we selected these for expression in yeast in addition to bacteria.

The AFP13 in the operons of these EATs that are toxic to both either have similarity to both phage tails and eukaryotic virus-like elements (i.e. inconclusive results), or are missing AFP13 outright in their operons (see last column in Table 1 that we added due to reviewers requests). Therefore, we cannot draw wide conclusions from this.

It is possible that the toxicity in *E. coli* is non-specific. However, EATs 2,4,6,8, and 9 were all tested in both *E. coli* and Yeast, and we only found toxicity in *E. coli*, demonstrating specificity (see Table S13 that we have added). We added this information to the description of “Experimental validation of novel four eCIS toxins that target eukaryotic cells” in the results section.

- *After the text: “We predicted new EATs that target eukaryotes based on their presence in the 3’ end of the eCIS operon, presence of eukaryotic domains within the encoded proteins, predicted enzymatic activity on eukaryote-specific molecules (such as actin), and similarity to known virulence factors.”, a reference with the table were these effectors are detailed should be added and colour code the selected ones (similar than before for antibacterial effectors).*

We added a reference to Table S13 of all tested genes

- *“Upon expression induction, we identified four EATs that efficiently killed yeast (Figures 6, S9).” Figure S9 does not correspond here and a comma should be added after “induction”*

Added a comma and changed wording.

- *The database is a great tool but the “40% Cluster Group” is not clear what does it refers to”*

The 40% Cluster Group is a way to cluster the accessory genes into families. Our assumption is that genes with $\geq 40\%$ identity (and $\geq 80\%$ coverage) have similar functionality, so we give them a cluster ID number to mark them as the same. An explanation is provided at the website and the Materials and Methods section (lines 579-582):

"eCISem contains information on "40% Cluster Group" of proteins that share $\geq 40\%$ identity and $\geq 80\%$ coverage based on CDHIT72 clustering. We assume these proteins, that their genes are part of eCIS operons, have similar functionality, so we give them a cluster ID number to mark them as the same."

- *In the discussion, the authors comment on the fact that there were already 7 EATs*

identified and that they have added 13 novel ones. Does their methodology allow the identification of the 7-known EATs? As a positive control of how much it covers

We indeed capture four of the seven from the literature as part of our predicted eCIS operons. We suppose that if our arbitrary boundaries were expanded, we would identify more of the effectors, as we have the known eCIS in our database.

Literature (genbank)	eCISem (IMG gene ID)
AAT48355.1 (afp18)	640051130
WP_015834234.1 (RRSP-Like Effector)	644888485
CAR67759.1 (A PVC effector)	644889759
CAE13983.1 (plu1690)	637463496

Here is an example of an effector we missed:

pne1 from Rocchi et al. 2019 (IMG gene 2512913831). The gene is found beyond the borders of our predicted eCIS operon (operon ID 1535) as this operon from *Pseudoalteromonas luteoviolacea* HI1 may have divided into two parts.

Regarding the bioinformatic prediction process – we used the common features from known EATs. For example: the fact that the EAT genes are located at the 3' end of the eCIS operon.

- *In the discussion, the authors stated: “The fact that most eCIS tail fibers matched both phage and eukaryotic-targeting viruses suggests they may have broad spectrum binding activity, and therefore broad targeting activity.” Whereas while reading the results section, they implied that only 18% matched phage or eukaryotic viruses and most didn’t match any of them. This point needs to be clarified.*

We have clarified this point by creating Figure S11, which shows all Afp13s in our search space, and whether they had a preferential hit against either phages or Eukaryotic viruses, had a similar score towards both, or had no hit whatsoever (no hits against either database. Further, Figure S12 shows each Afp13 gene that did have any blast hit on the X-axis, and on the Y-axis the bitscore of the best hit against the Eukaryotic-targeting viruses minus the bitscore of the best hit against tailed bacteriophages.

We changed the text at lines 187-192 to clarify the point and added a pointer to the new supplementary figures.

- *Figure 1B, genera should be italicised*

Changed genera in Figure 1B to italics.

- *The authors should be consistent and use either “fiber” or “fibre”, but always the same variant*

Changed all to consistently be spelled as “fiber”

- *In the text: “In some cases, we were able to experimentally identify immunity genes*

against that rescued bacteria from self-intoxication by their cognate toxins, supporting the toxin intended activity against bacteria". The word "against" is not necessary

Changed.

- *Figure reference in the text does not appear in order, for example, Figure S6 appear before Figure S5 in the main text, it is not the only one (Fig S8 before S7), double-check all of them*

Changed.

- *Table 1 – errata DUF4147 instead of DUF4157*

Changed.

- *Fig 5. B. XRE-like Antitoxin Protects From RES-like Toxin – no need to capitalise every first letter of every word*

Changed to lowercase.

- *In the text: "In one case, The toxin, EAT5, resembles RES (Figure 4b), an NADase that is accompanied by an antitoxin called Xre50,51." T from The should not be capitalised.*

Changed to lowercase.

- *The text: "Following our extensive bioinformatic analysis we were also interested in discovery of novel eCIS anti-eukaryotic toxins." Would read better like this: "we were also interested in discovering novel eCIS ..."*

Changed.

- *After the text: "EAT13 gene was tested only in yeast. These are predicted to act as an ADP-ribosyl transferase, a chitinase, a deaminase, and an actin cross-linker", the word "respectively" should be added*

Changed.

- *In the text: " We identified that some eCIS tail fibres that can help predict whether eCIS target bacteria or eukaryotes.", the second "that" after "fibres" should be eliminated*

Changed.

Thanks again for a very thorough review!

REVIEWERS' COMMENTS

Reviewer #1 (Remarks to the Author):

I am satisfied that the authors have addressed the issues raised in my first review and have correspondingly improved the paper. A few very minor issues:

- Generally the added supplemental figures have greatly expanded and improved the manuscript, however in some cases (ex. fig S23) the legends are quite brief and would benefit from more description.
- Please indicate in legend of Figure S21 if the data are from the biological replicates in the main figure. Also, the Y axis is Log, with 10^1 , 10^2 , etc as units, but the legend states log10.
- Line 900, "syntenny"

Reviewer #2 (Remarks to the Author):

I thank the authors for answering all my questions and taking on board all of my suggestions.

I just have one very minor request that I neglected to mention before - it would be good to have some clearer information on how the environment metadata info was obtained that was used as input to Scoary. Presumably this was downloaded along with the genomes from IMG? And I imagine the most detailed metadata is only available for a subset of genomes?

Reviewer #3 (Remarks to the Author):

The authors have addressed all my comments and criticisms very carefully and I do not have any further concerns. This is a great study that significantly increases our understanding of the eCIS and it has been a pleasure to review.

Patricia Bernal

Reviewer #1 (Remarks to the Author):

I am satisfied that the authors have addressed the issues raised in my first review and have correspondingly improved the paper. A few very minor issues:

Thank you for your review.

- Generally the added supplemental figures have greatly expanded and improved the manuscript, however in some cases (ex. fig S23) the legends are quite brief and would benefit from more description.

The legends have been expanded with more details.

- Please indicate in legend of Figure S21 if the data are from the biological replicates in the main figure. Also, the Y axis is Log, with 10^1 , 10^2 , etc as units, but the legend states log10.

Changed the Y-axis to properly match the description.

- Line 900, "syntenny"

Fixed.

Reviewer #2 (Remarks to the Author):

I thank the authors for answering all my questions and taking on board all of my suggestions.

Thank you for your detailed review and suggestions.

I just have one very minor request that I neglected to mention before - it would be good to have some clearer information on how the environment metadata info was obtained that was used as input to Scoary. Presumably this was downloaded along with the genomes from IMG? And I imagine the most detailed metadata is only available for a subset of genomes?

We indeed downloaded the metadata from the IMG database. There were fourteen metadata categories we considered (e.g. "Temperature range", "Ecosystem", etc.). Each category was filled in to various degrees by the groups that submitted the genomes to the IMG, which sometimes lead to missing information. By doing multiple analyses with different parameters, and by doing quality control by manually inspecting the reliability of our statistical results (Figure S8), we are confident in our results, despite the "real world noise" in the data.

Reviewer #3 (Remarks to the Author):

The authors have addressed all my comments and criticisms very carefully and I do not have any further concerns. This is a great study that significantly increases our understanding of the eCIS and it has been a pleasure to review.

Patricia Bernal

Thank you Patricia! We appreciate your close reading.